EMBO
Molecular Medicine

# Immune-enhancing neutrophils reprogrammed by subclinical low-dose endotoxin in cancer treatment

Yao Zhang [ID], Christina Lee, Shuo Geng, Jing Wang [ID], Udipta Bohara, Jacqueline Hou, Ziyue Yi & Liwu Li [ID] [✉]

## Abstract

**Despite the re-emergence of the pioneering "Coley's toxin" concept in anti-cancer immune therapies highlighted by check-point inhibitors and CAR-T approaches, fundamental mechanisms responsible for the immune-enhancing efficacy of low-dose "Coley's toxin" remain poorly understood. This study examines the novel reprogramming of immune-enhancing neutrophils by super-low dose endotoxin conducive for anti-cancer therapies. Through integrated analyses including scRNAseq and functional characterizations, we examined the efficacy of reprogrammed neutrophils in treating experimental cancer. We observed that neutrophils trained by super-low dose endotoxin adopt a potent immune-enhancing phenotype characterized by CD177$^{lo}$CD11b$^{lo}$CD80$^{hi}$CD40$^{hi}$Dectin2$^{hi}$. Both murine and human neutrophils trained by super-low dose endotoxin exhibit relieved suppression of adaptive T cells as compared to un-trained neutrophils. Functionally, neutrophils trained by super-low dose endotoxin can potently reduce tumor burden when transfused into recipient tumor-bearing mice. Mechanistically, Super-low dose endotoxin enables the generation of immune-enhancing neutrophils through activating STAT5 and reducing innate suppressor IRAK-M. Together, our data clarify the long-held mystery of "Coley's toxin" in rejuvenating anti-tumor immune defense, and provide a proof-of-concept in developing innate neutrophil-based anti-tumor therapeutics.**

**Keywords** Neutrophil Reprogramming; Immune-enhancement; Coley's Toxin; LPS; Cancer Treatment
**Subject Categories** Cancer; Immunology

## Introduction

The concept of cancer-immune therapies originated from the early studies related to "Coley's toxin", in which Dr. Coley employed the injection of low-dose bacterial extracts to induce low-grade fever and improve host broad-spectrum anti-tumor defense (Hoption Cann et al, 2003). Such practice laid a foundation for the recent emergence of immune-based tumor therapies. Although subclinical low dose endotoxin may elicit low-grade immune-enhancing effects, higher dose endotoxin challenges are known to induce innate exhaustion characterized by pathogenic inflammation and immune suppression (Pradhan et al, 2021), which may promote tumor progression. Such predicament halted the effective development of Coley's toxin-based anti-cancer therapies. Instead, the resurgence of modern day immune therapies evolved into largely harnessing the potential of adaptive immune therapies based on CAR-T approaches. The original concept of training innate immunity by low-dose microbial toxin has been largely ignored and poorly examined.

We have performed systems studies examining the dosage-dependent programming of innate immunity by bacterial endotoxin (lipopolysaccharides, LPS), and revealed complex dynamics of priming, tolerance and exhaustion depending upon the dosages of bacterial endotoxin (Pradhan et al, 2021; Yuan et al, 2016). While super-low dose LPS preferentially induces a low-grade inflammatory response from monocytes (Yuan et al, 2016), a higher dose LPS drastically causes tolerance and exhaustion of monocytes (Pradhan et al, 2021). Our findings potentially explain the lack of robust clinical effectiveness of the original Coley's toxin concept. In terms of the cellular component responsible for modulating tumor immune environments, recent studies increasingly reveal the important roles of neutrophils (Powell and Huttenlocher, 2016; Shen et al, 2014) (Gentles et al, 2015; Shen et al, 2014)). A recent clinical and translational study revealed that, instead of the engineered CAR-T cells, neutrophils are predominantly responsible for the broad-spectrum eradication of heterogeneous tumor cells in vivo (Hirschhorn et al, 2023). Intriguingly, neutrophils can exhibit either pro- or anti-tumor efficacies, potentially dependent on their distinct subsets and/or activation states (Sagiv et al, 2015; Shaul and Fridlender, 2019). Despite these important discoveries, differential polarizations of neutrophils by endotoxin and their contributions to the modulation of anti-tumor immunity are poorly understood.

Based on these intriguing clues, we herein examined whether super-low dose LPS may preferentially reprogram neutrophils into an immune-enhancing state, conducive for the treatment of cancer. We performed single-cell RNA sequencing (scRNAseq) of neutrophils trained by super-low dose LPS, and identified a reprogramming of immune-enhancing neutrophil cluster. Functionally, we performed adoptive transfer experiments demonstrating that the transfusion of super-low dose LPS trained neutrophils

Department of Biological Sciences, Virginia Tech, Blacksburg, VA 24061-0910, USA. ✉E-mail: lwli@vt.edu

can effectively reduce the tumor burden in recipient mice induced by azoxymethane (AOM)-dextran sulfate sodium salt (DSS) treatment. Super-low dose LPS trained neutrophils also relieved the suppression of adaptive T cells in the coculture assays as compared to naïve neutrophils. Mechanistically, we defined that super-low dose LPS selectively clears away the innate suppressor IRAK-M, initiates a sustained activation of STAT5, suppresses immune-suppressive STAT1/3, and reprograms neutrophils into an immune-enhanced state.

# Results

## Reprogramming of immune-enhancing neutrophils by super-low dose endotoxin

Emerging data reveal that LPS can dose-dependently modulate monocyte activation status, with prolonged high-dose LPS causing immune suppression and prolonged subclinical super-low dose LPS generating immune-enhancement (Pradhan et al, 2021; Yuan et al, 2016). In terms of neutrophils, the effects of high-dose LPS have been examined with elevated expression of immune-suppressive molecules such as PD-L1 as well as pathogenic inflammatory marker CD11b (Li et al, 2023b; Zhou et al, 2005) (Appendix Fig. S1). However, the effect of super-low dose LPS has not been well-characterized. Independent studies reveal that both murine and human neutrophils can be robustly maintained by ex vivo culture for several days in the presence of a very low-dose GM-CSF, without activating or compromising their activities (Guthridge et al, 2006; Kumar et al, 2022; Sun et al, 2018; Zhan et al, 2019). We therefore adopted such culture system to maintain the survival of purified neutrophils. We tested and validated that indeed low-dose (1 ng/ml) GM-CSF-maintained neutrophils can maintain robust survival within the 24 h culture period, with or without addition of LPS (100 pg/ml) (Appendix Fig. S2).

Cultured neutrophils were then subjected to scRNAseq analyses. As shown in Fig. 1A, control neutrophils cluster into three groups as we reported previously (Lin et al, 2022) (CD177$^{hi}$, CD177$^{intermediate}$, and CD177$^{lo}$/CD200R$^{hi}$). The cluster of CD177$^{intermediate}$ neutrophils appears to be derived from the CD177$^{hi}$ cluster (Appendix Fig. S3A) based on pseudo-time trajectory analysis, with both clusters share features of elevated inflammatory markers (CD177, CD11b). In contrast, the unique CD177$^{lo}$ cluster appears to be off a separate origin, and preferentially exhibits enhanced immune-enhancing gene markers (e.g., CD44, CD80, CD74, EHD1, Dectin2, and CD40), as well as reduced pathogenic inflammatory markers (CD177, CD11b). Upon overnight treatment with super-low dose LPS, neutrophils were preferentially projected into the unique CD177$^{lo}$ cluster, characterized by reduced expression of immune-suppressive/pathogenic inflammatory CD11b and elevated expression of immune-enhancing genes including CD44, CD80, CD74, EHD1, Dectin2, and CD40 (Fig. 1B; Appendix Fig. S3B). We further validated this intriguing observation at the protein level via flow cytometry. As shown in Fig. 1C and Appendix Fig. S4, neutrophils programmed by subclinical super-low dose LPS exhibited elevated expression of CD44, CD80, CD40, EHD1, Dectin2, CD200R, CD62L, and reduced expression of CD11b. This is in sharp contrast with neutrophils treated with higher dose LPS causing immune-suppression/pathogenic

inflammation, with elevated levels of CD11b and shredding of CD62L (Appendix Fig. S1) Our data reveal that neutrophils programmed by low-dose LPS may preferentially adopt an immune-enhancing, yet less pathogenic inflammatory/immune suppressive phenotype amenable for effective host anti-tumor immune defense, without eliciting adverse inflammatory damage.

## Reprogrammed neutrophils by super-low dose endotoxin enhance host anti-tumor defense

Our scRNAseq analyses with super-low dose LPS harkens back to the decade-old concept of "Coley's toxin" where Dr. Coley treated his cancer patients by injecting tiny amounts of microbial extracts (Hoption Cann et al, 2003). To further clarify this intriguing concept, we tested whether neutrophils precisely trained with subclinical dose LPS in vitro can be used to treat cancer in experimental animals. To test this, we used the azoxymethane (AOM)-dextran sulfate sodium salt (DSS) regimen which is a well-defined colorectal cancer model consistently used in our group (Zhang et al, 2020; Zhang et al, 2019). First, bone marrow neutrophils were isolated and primed with LPS (100 pg/ml) or PBS as control. Then, LPS or PBS-primed neutrophils were injected i.p. weekly to WT mice subjected to AOM-DSS challenge as described in the Method section (Fig. 2A). Transferred neutrophils could be detected in the circulation, inflamed colon, and other multiple tissues (Appendix Fig. S5A). As shown in Fig. 2B, mice receiving control neutrophils develop significant amount of colon tumors after AOM-DSS treatment, well within the statistical range of tumor loads we observed under the same condition in mice without any neutrophil infusion as we observed before (Zhang et al, 2019) (Appendix Fig. S5B). In contrast, mice receiving LPS-primed neutrophils exhibited a significant reduction in the tumor loads. In particular, the average numbers of microscopic (<2 mm diameter) and macroscopic (≥2 mm diameter) polyps in the mice receiving LPS-primed neutrophils were reduced by 50–65% compared with that in the mice receiving PBS-primed neutrophils. Moreover, the mice receiving LPS-primed neutrophils displayed reduced weight loss and much lower disease scores (Fig. 2C,D) during the course of AOM-DSS treatment. In addition, consistent with less severe colonic symptom at the end of AOM-DSS regimen, histologically, H&E staining illustrated that transfusion with LPS-primed neutrophils reduced colon inflammation and alleviated epithelial structure (Fig. 2E). Flow cytometry analysis further confirmed a reduction of infiltrating neutrophils in the colon, liver, spleen and mesenteric lymph nodes in recipients receiving LPS-primed neutrophils compared to recipients receiving control neutrophils (Appendix Fig. S6). Moreover, we stained colon sections for Ki67, a proliferation marker associated with poor clinical outcome in the patient with colorectal cancer (Luo et al, 2019). The control mice showed pervasive Ki67 staining, while mice receiving LPS-primed neutrophils exhibited significantly reduced Ki67 positive staining in colon tissues (Fig. 2F). In addition, the expression of β-catenin, another independent maker for tumorigenesis, was also reduced in colon tissues of animals transfused with LPS-primed neutrophils (Fig. 2G). Our data indicate that transfusion of LPS-primed neutrophils enhances anti-tumor activities against AOM-DSS induced colon tumorigenesis.

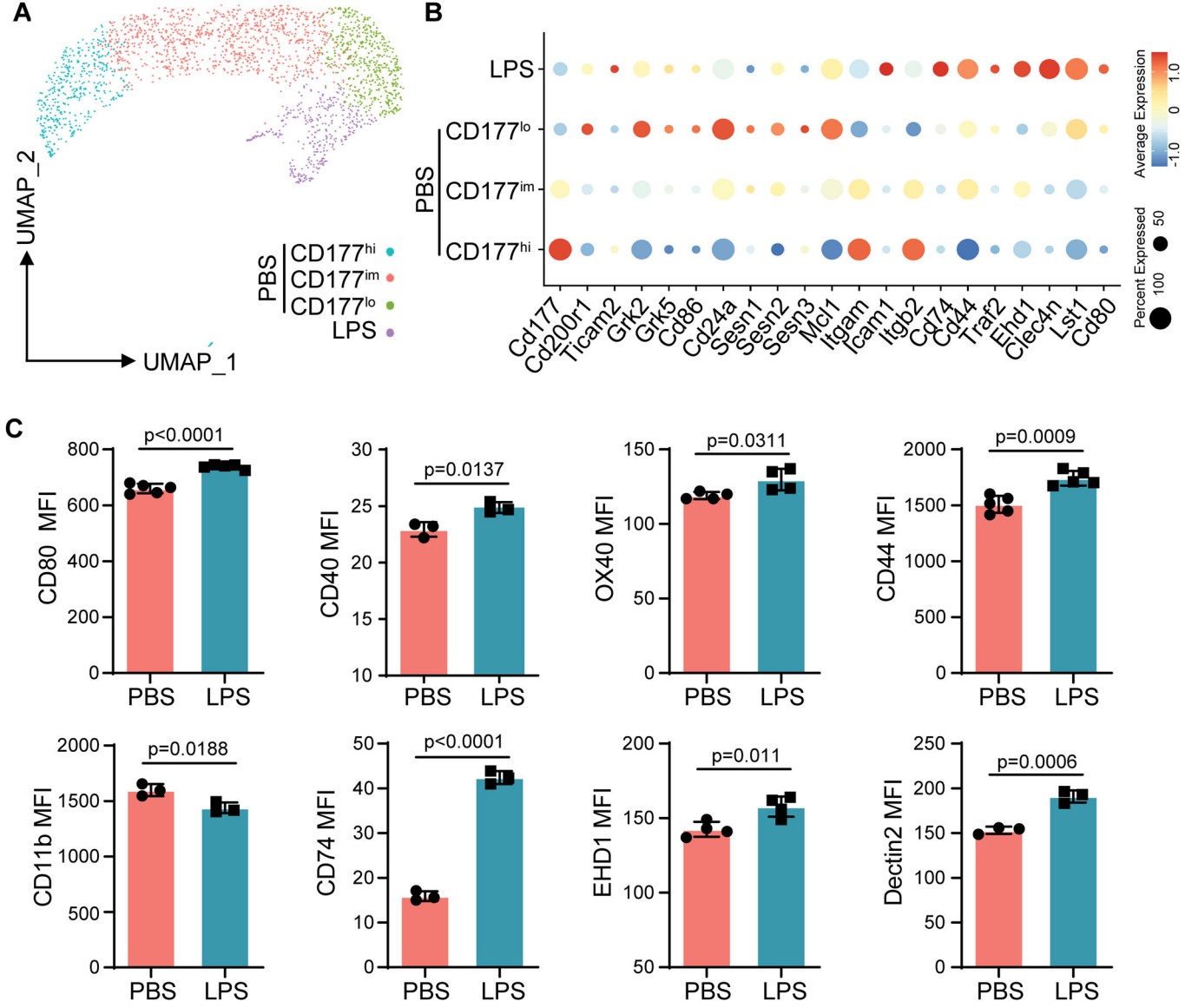

**Figure 1. Reprogramming of immune-enhancing neutrophils by super-low dose endotoxin.**

(A) The UMAP diagram of scRNA-Seq data from purified neutrophils treated with 100 pg/ml LPS or PBS as control. Control neutrophils cluster into three clusters: CD177high (CD177hi), CD177intermediate (CD177im), and CD177low (CD177lo). (B) Dot plots of relative gene expression levels in different clusters of neutrophils. (C) Bone marrow neutrophils were treated with 100 pg/ml of LPS or PBS as control overnight, then subjected to flow cytometry analysis. Mean fluorescence intensity (MFI) of selected proteins on CD11b⁺Ly6G⁺ neutrophils were analyzed. n ≥ 3 (biological replicates). Data are represented as means ± SD. Significance was calculated by unpaired two-tailed student t test. Source data are available online for this figure.

## Reprogrammed neutrophils by super-low dose endotoxin enhance anti-tumor immune environment in vivo

Next, we examined the T cell status after transfusion of LPS-primed neutrophils. Significantly higher cell counts of CD4 positive and CD8 positive T cells were observed in the spleen (Fig. 3A), mesenteric lymph nodes (Fig. 3B), as well as colon tissues (Appendix Fig. S7) of the mice receiving LPS-primed neutrophils compared with the mice receiving PBS-primed neutrophils during AOM-DSS treatment. The percentage of Ki67 + CD4 + T cells was much higher in the group of LPS-primed neutrophil recipients (Fig. 3C). Moreover, the reduction of Foxp3 + CD4+ Treg cells and

CD122 + CD8+ Treg cells were observed in both spleen and lymph modes from the group of LPS-primed neutrophil recipients compared with the control group (Fig. 3D; Appendix Fig. S8A). Furthermore, not only the cell number but also the activation status of T cells was induced by weekly transfusion of LPS-primed neutrophils. In particular, the percentage of CD107a+ in CD8 + T cells was much higher in LPS-primed neutrophil recipients, and the expression of granzyme B and interferon γ in CD8 + T cells were significantly increased as well (Fig. 3E,F; Appendix Fig. S8B). On the other hand, the expression of PD-1 and Tim3, immune checkpoint molecules, were remarkably decreased in the CD4+ and CD8 + T cells in the spleen and lymph nodes

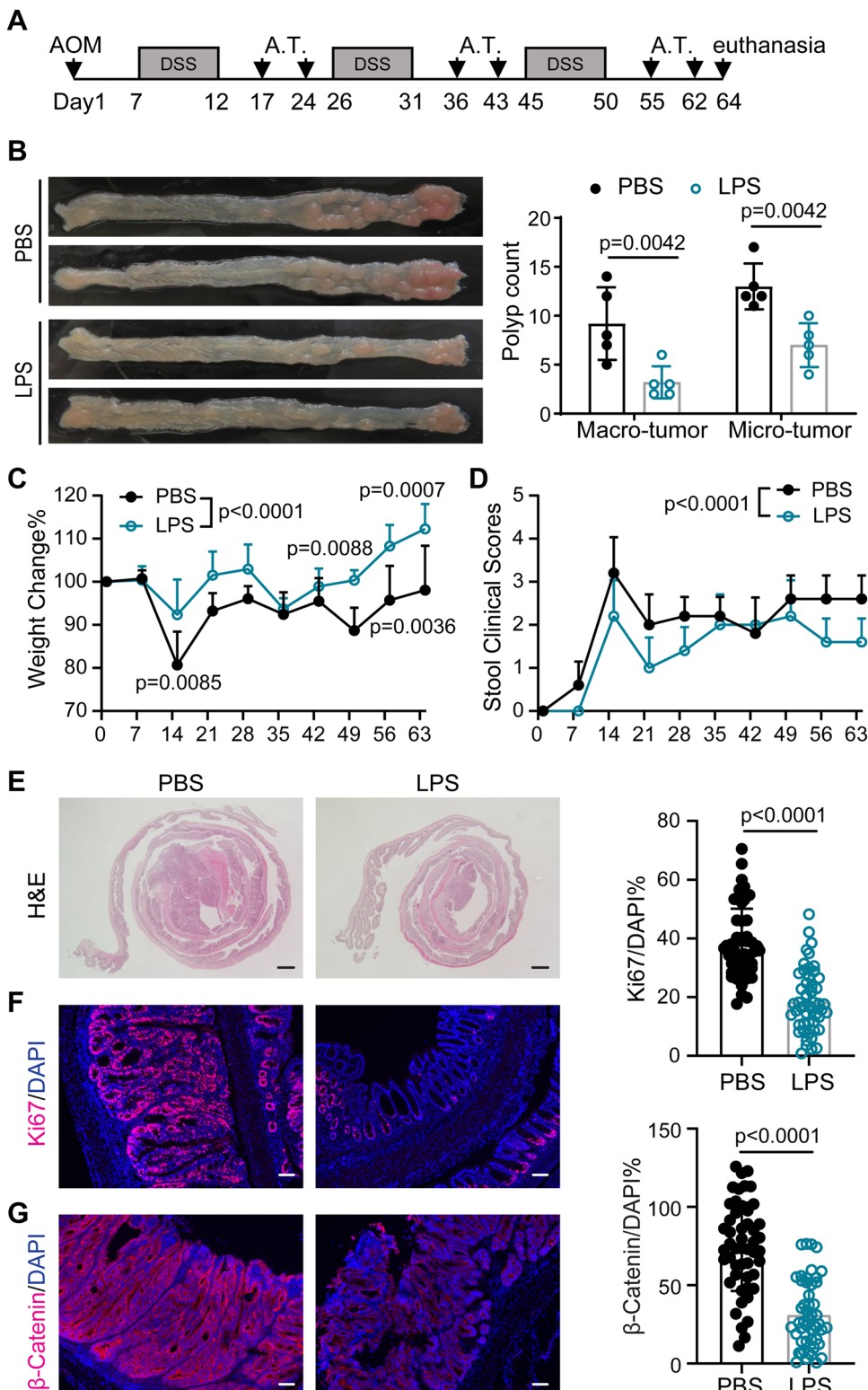

from LPS-primed neutrophil recipients compared with the control group (Fig. 3G; Appendix Fig. S8C).

NK cells also play an essential role in anti-tumor immunity, and neutrophils also affect NK cell activities (Palano et al, 2021). We next examine whether transfusion of LPS-primed neutrophils

affects NK cells in vivo. First, the cell counts of NK cells were significantly and persistently higher in the spleen (Fig. 4A) and mesenteric lymph nodes (Fig. 4B) from the mice receiving LPS-primed neutrophils. Moreover, the expression of NKG2A, an immune check point, was also down-regulated on the NK cells

**Figure 2.  LPS-primed neutrophils slow down colitis-associated colon cancer progression.**

(A) Schematic experimental design for adoptive transfer (A.T.) of neutrophils in the AOM-DSS-induced colitis-associated colon cancer model. AOM-DSS treated WT mice were given with LPS or PBS-primed neutrophils via intravenous injection. (B) Representative images of colons from the mice receiving PBS or LPS-primed neutrophils after AOM-DSS treatment. Diameter of tumors greater than or equal to 2 mm defined as "macro" tumor; diameter of tumors less than 2 mm defined as "micro" tumor. $n = 5$ (biological replicates). Data are represented as means ± SD. Significance was calculated by two-way ANOVA with Sidak post hoc test. (C) Body weight changes during AOM-DSS treatment. $n = 5$ (biological replicates). Data are represented as means ± SD. Significance was calculated by two-way ANOVA with Sidak post hoc test. (D) Stool clinical scores including stool consistency and bleeding. $n = 5$ (biological replicates). Data are represented as means ± SD. Significance was calculated by two-way ANOVA with Sidak post hoc test. (E) Representative images of H&E-stained colon sections. Scale bars: 1 mm. (F, G) Immunofluorescent staining analysis of Ki67 and β-catenin within colon sections. Scale bars: 100 μm. $n = 50$ fields from three independent animal specimens. Data are represented as means ± SD. Significance was calculated by unpaired two-tailed student $t$ test. Source data are available online for this figure.

from the mice receiving LPS-primed neutrophils (Fig. 4C,D). Meanwhile, the cytokine production of NK cells in LPS-primed neutrophil recipients was enhanced, as reflected with the higher level of CD107a on NK cells (Fig. 4E,F), as well as higher percentage of granzyme B positive cells in the mice receiving LPS-primed neutrophils (Fig. 4G,H). In addition, we also observed that the percentages of colonic NK cells from mice receiving LPS-primed neutrophils were higher than those from control mice after AOM-DSS treatment (Appendix Fig. S7E), and CD107a expression level was also remarkably upregulated.

Consistent with elevated activation of T cells and NK cells, enhanced anti-tumor environments were detected in the colon, supported by the significant increase of granzyme B and interferon γ levels in the colon of LPS-primed neutrophil recipients. (Appendix Fig. S7F).

## Reprogrammed neutrophils by super-low dose endotoxin relieve adaptive immune suppression in vitro

To further explore whether super-low dose of LPS priming could enhance T cell function, we conducted in vitro coculture assays. The primed neutrophils were mixed with CFSE-labeled allogeneic T cells in CD3-coated plate for 72 h as described in the Method section based on our previous optimization (Zhang et al, 2019). PBS-treated neutrophils showed typical immunosuppressive phenotype, as evidenced from reduced T cell proliferation with the addition of neutrophils (Fig. 5A,B). However, when co-cultured with LPS-primed neutrophils, the proliferation of CD4 and CD8 T cells was markedly augmented and the immunosuppressive effect was partially relieved. The similar enhanced effects on T cell proliferation by LPS-primed neutrophils were also observed in the syngeneic murine T cell and neutrophil coculture system, and human T cell and neutrophil coculture system (Appendix Fig. 9A,B). Then we further explored the effect of neutrophils on T cell activation and cytokine production. Down-regulation of CD62L was observed in the T cells co-cultured with LPS-primed neutrophils, compared with the T cells co-cultured with PBS-primed neutrophils (Fig. 5C,D). Moreover, when co-cultured with LPS-primed neutrophils, CD8 + T cells exhibited significantly elevated activation status, supported by the higher levels of CD69 and CD107a as well as reduced PD-1 expression (Fig. 5E). In addition, the concentrations of Granzyme B and IFNγ were increased in the coculture with LPS-primed neutrophils (Fig. 5F). These data further confirm that T cell activation was enhanced when co-cultured with LPS-primed neutrophils as compared to control neutrophils, and the immune suppression caused by control neutrophils was alleviated.

## Reprogrammed neutrophils enhance NK cell activity in vitro

Not only did T cells interact with neutrophils, but also NK cell response was affected by neutrophils (Fig. 4). Therefore, we next examined the direct interaction between NK cells and neutrophils in vitro. Splenic NK cells were isolated and co-cultured with either PBS- or LPS-primed neutrophils in the presence of IL2. The proliferation of NK cells was suppressed by addition of neutrophils. However, compared with PBS-primed neutrophils, LPS-primed neutrophils partially released the suppression of NK cell proliferation (Fig. 6A,B). NKG2A is known as an immune checkpoint of NK cells. The percentage of NKG2A^hi NK cells was significantly reduced when NK cells were co-cultured with LPS-primed neutrophils (Fig. 6C). Next, the function of NK cells in the presence of neutrophils was tested. The secretion of granzyme B and IFNγ were elevated when NK cells were co-cultured with LPS-primed neutrophils as compared to PBS-primed neutrophils (Fig. 6D). The killing activities of NK cells to target YAC-1 cells were evaluated by both colorimetric measurement of LDH level in the culture medium and flow cytometry. Higher levels of LDH were detected in the medium with LPS-primed neutrophils than the controls (Fig. 6E). Furthermore, higher killing rates were consistently observed by counting the live fluorescence-labeled YAC-1 cells via flow cytometry (Fig. 6F).

Collectively, these data suggest that subclinical super-low dose LPS switch neutrophils to an immune-enhancing state, relieving the immunosuppressive effects of neutrophils on both T cell and NK cell functions in vitro.

## Super-low dose endotoxin reprograms neutrophils through activating STAT5 and reducing IRAK-M

We then examined the molecular mechanisms responsible for the immune-enhancing effects of super-low dose LPS. As shown with the monocyte studies, monocytes treated with super-low dose LPS initiate an activation of STAT5 as well as proteasome activation leading to the degradation of immune-suppressors such as IRAK-M (Geng et al, 2021; Yi et al, 2023). We then tested whether super-low dose LPS may similarly activate the program of STAT5 and immune-proteasome in neutrophils. As shown in Fig. 7A, neutrophils treated with super-low dose LPS exhibit elevated levels of phospho-STAT5, likely responsible for the sustained expression of immune-enhancing mediators. In the meantime, super-low dose LPS priming significantly elevated the levels of proteasome subunit PSMD10 (Fig. 7B). We then examined the levels of IRAK-M, a known immune suppressor in innate leukocytes (Kobayashi et al,

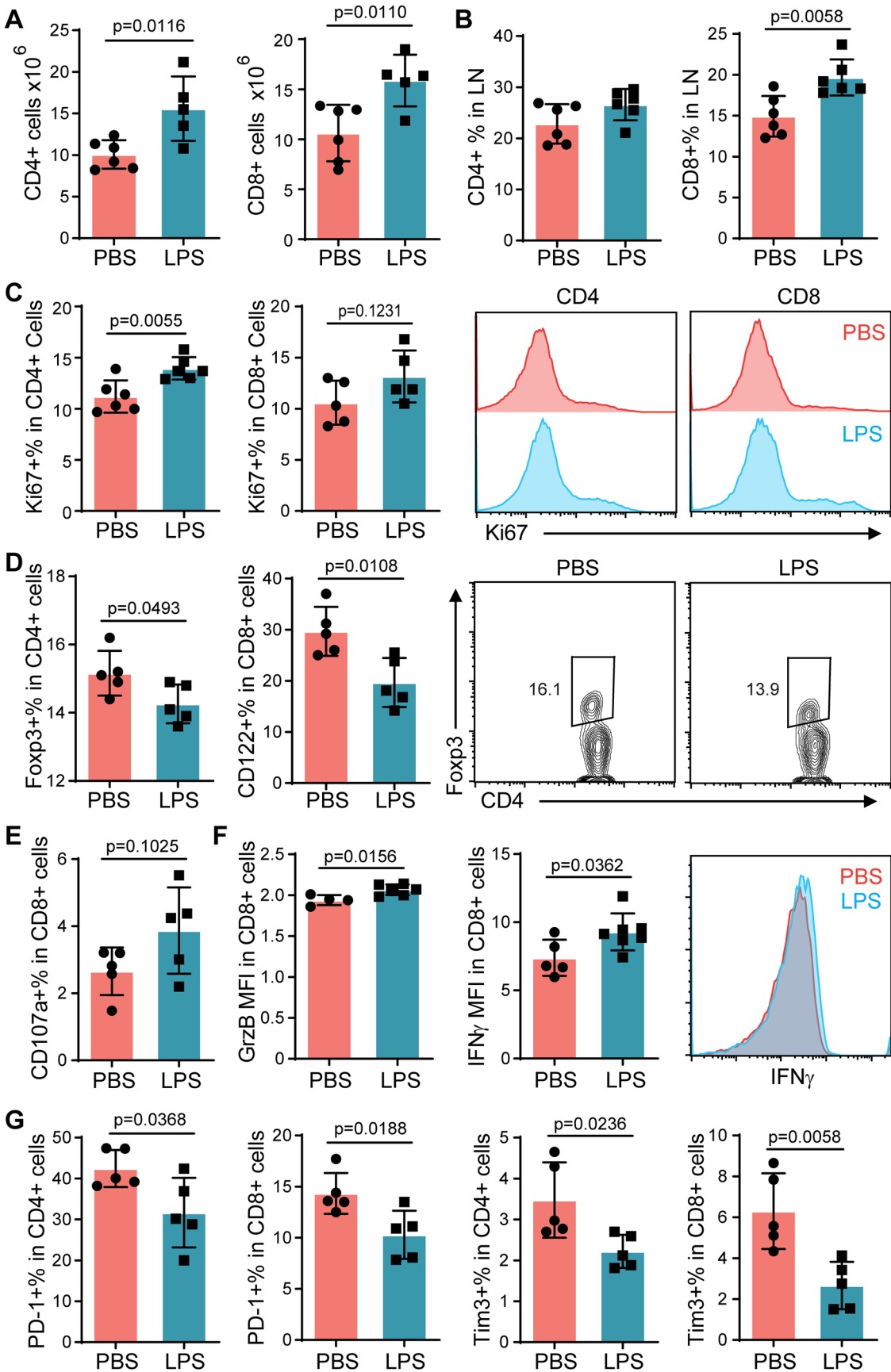

◄ **Figure 3. Enhanced T cell activities by transfusion with LPS-primed neutrophils in vivo.**

T cells from AOM-DSS challenged mice with adoptive transfer of neutrophils were analyzed by flow cytometry. (A) Cell counts of CD4+ and CD8 + T cells in the spleen. (B) The percentages of CD4+ and CD8 + T cells in the mesenteric lymph nodes. (C) Ki67 expression in splenic T cells. (D) Foxp3 expression in splenic CD4 + T cells and CD122 expression in splenic CD8 + T cells. (E) CD107a expression in splenic CD8 + T cells. (F) Splenocytes lymphocytes isolated from the experimental mice were cultured in RPMI completed medium supplemented with Golgi blocker for 4 h. Then IFNγ and Granzyme B (GrzB) expression were examined via flow cytometry. (G) PD-1 and Tim3 expression in splenic T cells. Data information: Data (A–G) are represented as means ± SD. $n \geq 5$ (biological replicates). Significance was calculated by unpaired two-tailed student $t$ test. Source data are available online for this figure.

2002), and observed a significant reduction of IRAK-M in LPS-primed neutrophils (Fig. 7C). IRAK-M not only suppresses the immune-enhancing signals, but was also shown to be involved in the activation of immune suppressors such as CD11b (Gong et al, 2017). We previously demonstrated that IRAK-M deficient neutrophils have significantly reduced expression of CD11b (Zhang et al, 2020). Complementing these earlier observations, we demonstrated that neutrophils treated with super-low dose LPS exhibited reduced levels of KLF4 (Fig. 7D), a known transcription factor involved in CD11b expression (Feinberg et al, 2007). KLF4 was known to be under the control of STAT3/1 (Chen et al, 2002; Ganguly et al, 2021; Luo et al, 2024), and collectively contribute to the induction of CD11b. An independent study with another cellular system reported that IRAK-M can closely interact with STAT3 and potently stabilize STAT3 (Kesselring et al, 2016). Consistent with such finding, we previously reported a reduction of STAT3 and STAT1 in IRAK-M deficient neutrophils (Zhang et al, 2020). To further validate such mechanism in neutrophils, we performed additional experiments testing the interaction of IRAK-M with both STAT1 and STAT3 in neutrophils. Indeed, through co-immunoprecipitation assays, we detected an interaction of IRAK-M with both STAT1 and STAT3 (Fig. 7E,F). Further, we performed nuclear isolation and tested the nuclear levels of STAT1/3. As shown in Fig. 7G,H, LPS priming significantly reduced STAT3 and STAT1 in the nuclear fraction, validating the reduction of nuclear levels of STAT3 and STAT1 by LPS. Collectively, our data suggest that the removal of IRAK-M-related suppressors may enable the establishment of immune-enhancing neutrophils by super-low dose LPS.

## Discussion

Our data clarify the novel reprogramming potential of super-low dose endotoxin on neutrophils, shifting them from an immune-suppressive state to an immune-enhancing phenotype. Our findings clarify the mystique surrounding the original concept of "Coley's toxin" in experimenting with low levels of bacterial extract for improving anti-tumor immune responses (Coley, 1910). While neutrophils exhibit both immune-suppressive and immune-enhancing potentials, our data reveal that the fate of neutrophils bifurcates depending upon the signal strength of endotoxin, with super-low dose endotoxin preferentially promoting the immune-enhancing polarization. Our conclusion is not only supported by in vitro characterization of reprogrammed neutrophils, but also by in vivo proof-of-principle evidence showing enhanced anti-tumor defense in recipient mice receiving reprogrammed neutrophils with super-low dose endotoxin.

Our findings expand emerging studies that reveal a unique aspect of innate immune memory dynamics, in which innate leukocytes can discern not only the biochemical natures of danger signals, but more importantly the signal strength and duration (Geng et al, 2022). We previously reported that super-low dose LPS preferentially induces a low-grade inflammatory state in monocytes (Yuan et al, 2016), while a prolonged challenge with higher septic dose LPS yield an exhausted state with immune-suppressive characteristics (Pradhan et al, 2021). Our current data reveal a similar dynamic in neutrophils, and identify the immune-enhancing property of super-low dose endotoxin. Our data clarify a long-held puzzle with the early observation of "Coley's toxin" in treating cancer patients (Hoption Cann et al, 2003), in which the administration of bacterial extracts into human cancer patients led to variable clinical manifestations. Initial practices involved carefully dosing a tiny amount of bacterial extracts that was just enough to barely induce a systemic low-fever response from the patients. Due to the inevitable variations, the Coley's toxin was largely ignored in favor of more reproducible radiation- and chemo-therapies in the last decade. Even with the new dawn of immune therapies and constant referral of "Coley's toxin" as the foundation for the emergence of immune therapies (Hoption Cann et al, 2003), limited efforts were paid to evaluate the potential of innate leukocyte-based therapies. Rather, the main stream of cancer-immune therapies has been related to adaptive immunity coupled with check-point inhibition (Markman and Shiao, 2015). However, adaptive immune-based therapies suffer from the caveat of tumor evasion due to high mutation rates, similar to the draw-backs related to the vaccine approach targeting emerging pathogens.

Revisiting innate-based cancer therapies will hold the potential of generating effective and broad-spectrum anti-cancer therapies. Emerging studies reveal that innate leukocytes such as neutrophils are highly enriched in solid tumor tissues, adopting a pro-tumor phenotype (Wu et al, 2020). neutrophils promote tumor growth by serving as suppressors of adaptive immune cells through immune-suppressive/pathogenic inflammatory mediators including PD-L1, CD11b, and ROS (Cemerski et al, 2002; Varga et al, 2007; Wang et al, 2017), as well as assisting tumor cell growth and metastasis through CD11b-mediated swarming/aggregation with cancer cells (Li et al, 2023a). On the other hand, emerging studies reveal that neutrophils are also responsible for enhancing the broad-spectrum eradication of heterogeneous cancer cells that evade CAR-T therapies (Hirschhorn et al, 2023). Consistent with emerging single-cell sequencing analyses (Lin et al, 2022; Wigerblad et al, 2022), our data also reveal at least three distinct subsets of neutrophils with varying levels of CD177 and CD11b. The CD177[lo] population have reduced expression of immune-suppressive/pathogenic inflammatory markers such as CD11b, and preferentially express immune-enhancing mediators including CD80, CD86, CD40, Dectin2, and EHD1. Our data further reveal that super-low dose endotoxin preferentially expands the CD177[lo]

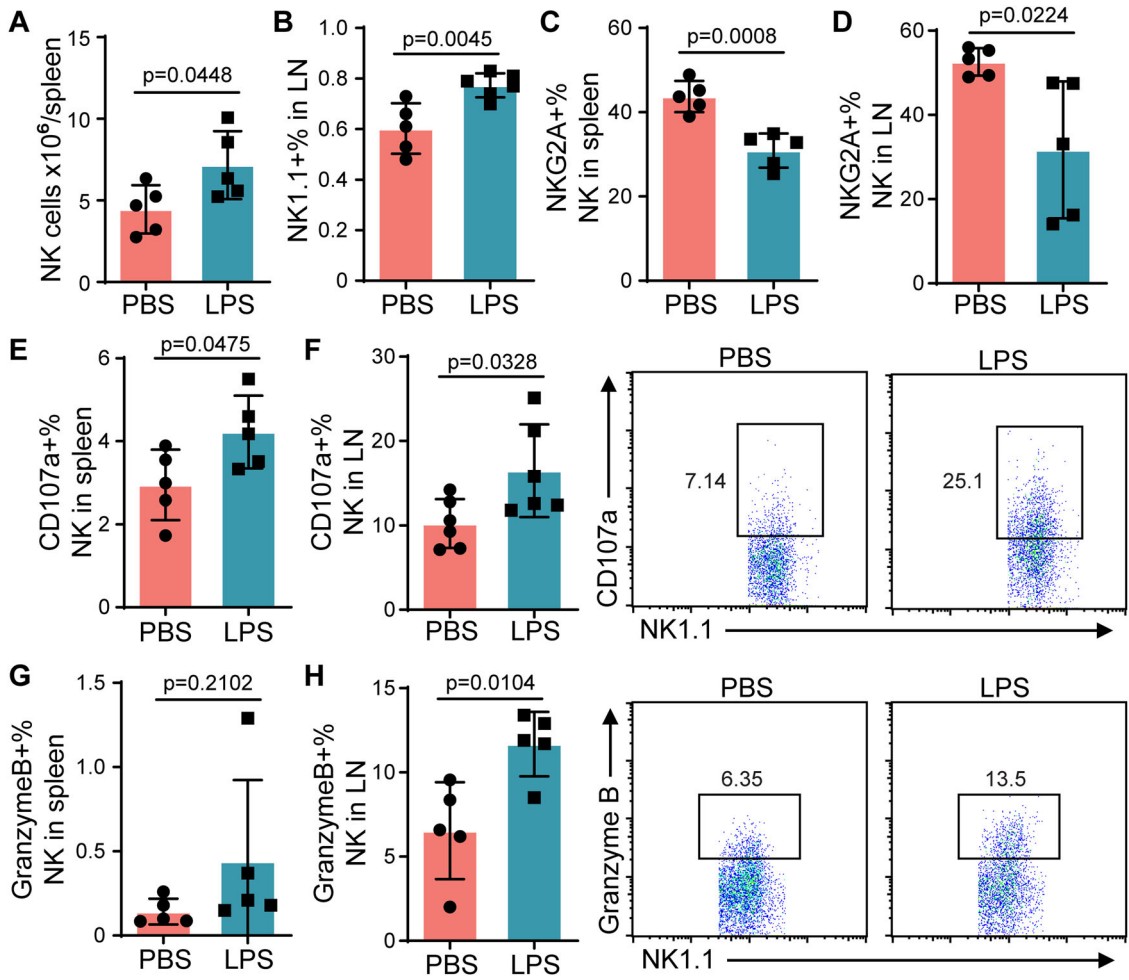

**Figure 4. Enhanced NK cell activities by transfusion with LPS-primed neutrophils in vivo.**

NK cells from AOM-DSS-challenged mice with adoptive transfer of neutrophils were analyzed via flow cytometry. **(A)** Cell counts of NK cells in the spleen. **(B)** The percentages of NK cells in the mesenteric lymph nodes. **(C)** NKG2A expression in NK cells of the spleen. **(D)** NKG2A expression in NK cells of the mesenteric lymph nodes. **(E)** CD107a expression in NK cells of the spleen. **(F)** CD107a expression in NK cells of the mesenteric lymph nodes. **(G)** Granzyme B expression in NK cells of the spleen. **(H)** Granzyme B expression in NK cells of the mesenteric lymph nodes. Data information: Data **(A–F)** are represented as means ± SD. $n \geq 5$ (biological replicates). Significance was calculated by unpaired two-tailed student $t$ test. Source data are available online for this figure.

immune-enhancing population. While this work was under revision, an independent basic and clinical study revealed that an immune-enhancing subset of neutrophils with a potent anti-tumor effect is conserved both in murine and human system, with a key signature of elevated Ly6A/E (Benguigui et al, 2024). We further validated that Ly6A/E were uniquely and abundantly expressed, preferentially within our CD177$^{lo}$ immune-enhancing neutrophils (Appendix Fig. S10A,B). Consistently, Ly6A/E expression can be robustly maintained by low-dose LPS, and drastically reduced upon high-dose LPS treatment (Appendix Fig. S10B,C). Our in vivo experiments validate the potent immune-enhancing potential of reprogrammed neutrophils by low-dose LPS in reducing experimental tumorigenesis.

Our study also reveals the potential molecular mechanisms involved in the immune-enhancing program of neutrophils, by selectively removing innate suppressor IRAK-M. This is consistent with several independent reports that genetic deletion of IRAK-M renders enhanced anti-tumor immune environment in reducing tumorigenesis ranging from lung cancer to colon cancer (Kesselring et al, 2016; Standiford et al, 2011). Neutrophils harvested from IRAK-M deficient mice share a similar immune-enhanced phenotype with reduced CD11b and enhanced CD80 (Zhang et al, 2020). The suppression of IRAK-M likely involves the activation of immune-proteasome which was independently reported to be activated during the initial response of innate leukocytes to challenges (Kammerl and Meiners, 2016; Qureshi et al, 2005). However, future in-depth studies are needed to further define the underlying mechanisms.

Overall, our study clarifies the mechanisms responsible for neutrophil reprogramming by super-low dose LPS and highlights its potential for enhancing anti-tumor immune responses. Our findings provide a basis for future mechanistic studies further characterizing the intriguing dynamics of neutrophils as well as clinical studies exploring the therapeutic applications of

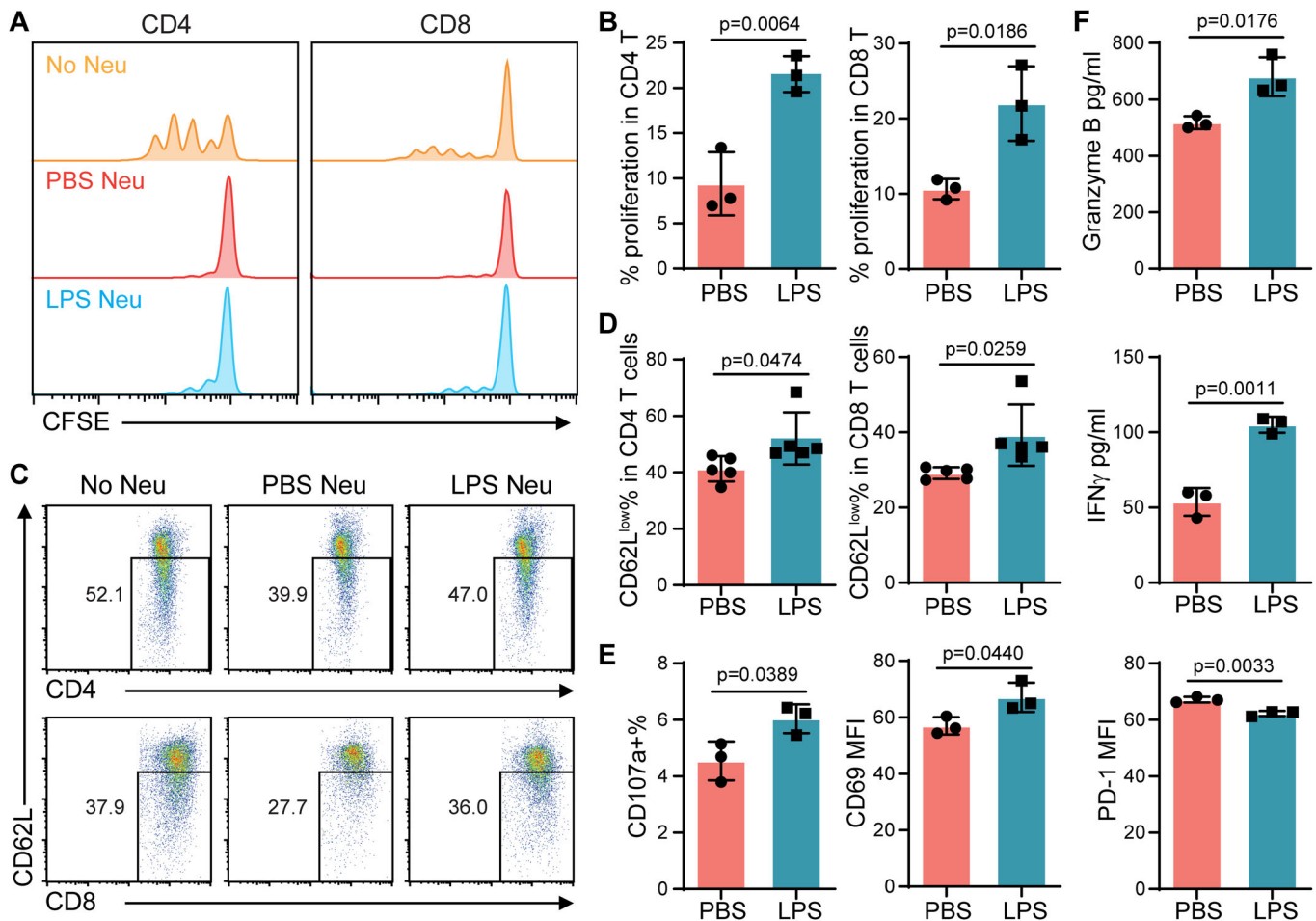

**Figure 5. Reprogrammed neutrophils by super low-dose LPS increase T cell proliferation and activation in vitro.**

(A) CFSE-labeled T cells were co-cultured with LPS-primed neutrophils for 72 h. Representative results of CFSE signal gated on CD4+ or CD8 + T cells are shown. (B) Quantification of the percentages of proliferation in CD4+ or CD8 + T cells. $n = 3$. (C) Representative plots of CD62L expression on CD4+ or CD8 + T cells after 3 days coculture. (D) Quantification of CD62L$^{low}$ percentages in CD4+ or CD8 + T cells. $n = 5$. (E) CD107a, CD69, and PD-1 expression in CD8 + T cells after three days coculture. $n = 3$. (F) IFNy and Granzyme B levels in conditioned medium. $n = 3$. Data information: Data (B, D–F) are represented as means ± SD of independent biological replicates ($n$). Significance was calculated by unpaired two-tailed student $t$ test. Source data are available online for this figure.

neutrophil-based cancer immunotherapies. However, our current study remains as a basic mechanistic study, potentially revealing a novel aspect of neutrophil dynamics amenable for future therapeutic exploration. Several key studies are needed to further translate its therapeutic usage. First, although programming by low-dose LPS may explain the mechanistic principle of "Coley's Toxin", the practical usage of human neutrophils trained by low-dose LPS is limited. Alternative and chemically well-defined training agent should be considered in more controlled and robust programming of immune-enhancing neutrophils for clinical usage. Second, pharmacodynamics and pharmacokinetics of programmed neutrophils will need to be systematically determined. Third, additional purification and optimization of neutrophil survival; storage and shipment will need to be carefully evaluated for neutrophil-based practical usage. Forth, temporal and spatial interaction of infused neutrophils with neighboring cells and tumor tissues should also be further extensively examined with

independent tumor models. Nevertheless, our current study as well as emerging studies clearly suggest the untapped potential of innate neutrophils, in providing another arsenal in the evolving field of innate immune therapy.

# Methods

## Mice

Wild type (WT) C57BL/6 (strain no. 000664) mice were originally from Jackson laboratory and bred and maintained in the animal facility at Virginia Tech in accordance to approved Animal Care and Use Committee protocol (reference no. 17193 and 20007). All littermate mice (both male and female) were 8–10 weeks of age and 25–30 g weight when experiments were initiated. Mice were randomly assigned to each group of treatment. Investigators were

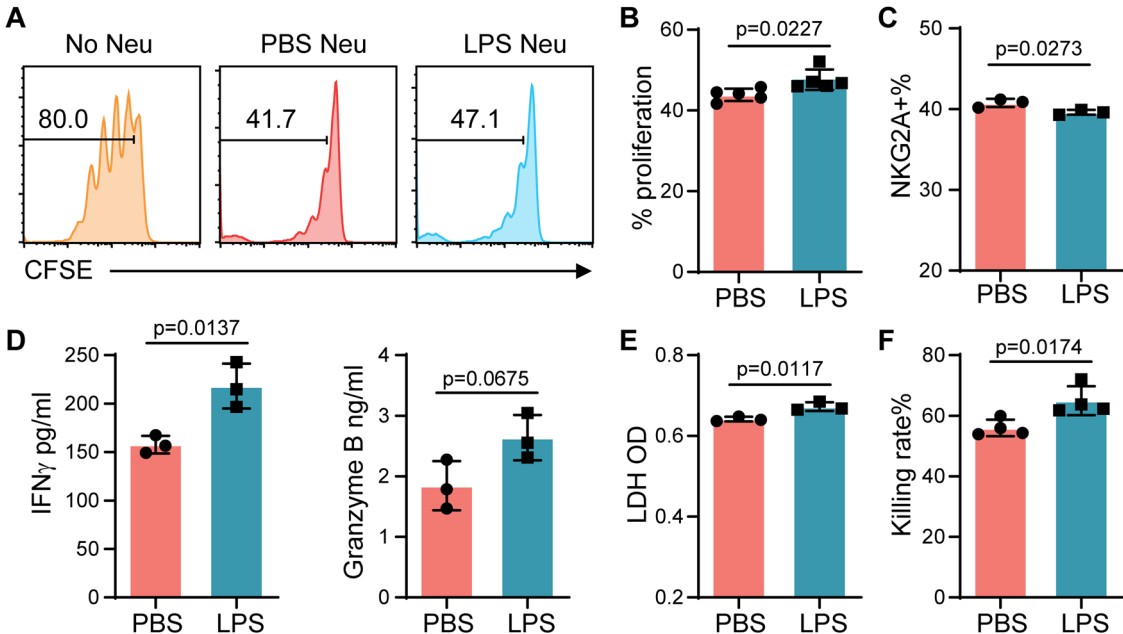

**Figure 6. Reprogrammed neutrophils by super low-dose LPS increase NK cell proliferation and functional ability in vitro.**

(A) CFSE-labeled NK cells were co-cultured with LPS-primed neutrophils for four days. Representative results of CFSE signal gated on NK1.1 + CD3- NK cells are shown. (B) Quantification of the percentages of proliferation in NK cells. $n = 5$. (C) NKG2A expression in NK cells after coculture. $n = 3$. (D) IFNγ and Granzyme B levels in conditioned medium. $n = 3$. (E) LDH levels in conditioned medium from NK cell killing assay. $n = 3$. (F) YAC-1 cells were used as target cells to evaluate NK cytotoxicity as mentioned in the Methods section. Killing rates were determined by counting remaining live target cells. $n = 4$. Data information: Data (**B–F**) are represented as means ± SD of independent biological replicates ($n$). Significance was calculated by unpaired two-tailed student $t$ test. Source data are available online for this figure.

not blinded. The study was performed in full compliance with the regulation set forth by the Virginia Tech IRB (Institutional Review Board) as well as the IACUC (Institutional Animal Care and Usage Committee).

## Human blood samples

This study belongs to non-human subject research. De-identified peripheral blood samples from healthy donors with no identifying information were commercially purchased from Research Blood Components, Boston, Massachusetts. Handling of human-exempt cells were performed in full compliance of the Virginia Tech IRB (Institutional Review Board).

## Experimental design

WT mice received a single intraperitoneal injection of azoxy-methane (AOM; MilliporeSigma, catalog no. A5486) at a dose of 10 mg/kg body weight. A week after AOM injection, the mice were given three cycles of 2% dextran sulfate sodium salt (DSS, MP Biomedicals, catalog no. 216011080) for 5 days followed by 14 days of normal drinking water. To prime neutrophils in vitro, bone marrow neutrophils from WT mice were purified (>90% confirmed by flow cytometry) using EasySep™ Mouse Neutrophil Enrichment Kit (Stem Cell, catalog no. 19762), according to the manufacturer's instruction. Purified neutrophils were cultured in RPMI completed medium (10% fetal bovine serum, 2 mM L-glutamine, 10 mM HEPES, 1% penicillin/streptomycin) supplemented with 1 ng/ml GM-CSF (Peprotech, catalog no. 315-03) and treated with LPS

(100 pg/ml; MilliporeSigma, catalog no. L2630) or PBS as control overnight. Then, the neutrophils were harvested and resuspended in PBS. Recipient WT mice were transfused twice (post DSS day 5 and day 12) per DSS-resting cycle through intravenous injection with $2.5 \times 10^6$ WT neutrophils. After the last water cycle mice were sacrificed and tissues were harvested for further analysis. Body weight, stool consistency, and bleeding were measured as parts of clinical score (score 0-4, with higher score corresponding to worse condition). Depending on the size, polyp formation was classified as macro-polyp (equal to or greater than 2 mm) and micro-polyp (less than 2 mm). Independent experiments of AOM-DSS induced colorectal tumorigenesis were conducted more than 3 times, and for each experiment there were at least 5 mice in each group.

## Single-cell RNAseq, cell clustering, and analyses

Cultured neutrophils were used for single-cell sequencing and analyses as described previously (Lee et al, 2021) with minor modifications. Briefly, Ly6G$^+$ positive neutrophils were purified from bone marrow via flow cytometry-based selection. They were then cultured with either PBS, or 100 pg/ml LPS overnight in completed RPMI medium supplemented with 1 ng/ml GM-CSF. Cultured cells were processed to prepare libraries using the 10X Genomics Chromium Single Cell 3' v3 Kit. Following 12 cycles of amplification, the qualities of cDNA samples were verified via Bioanalyzer and sequenced on Illumina® HiSeq platform, with the paired-end read length of 150 bp at each end plus 8-bp i7 index. scRNAseq data were analyzed by Seurat (version 3.1.2) in R as we described (Lee et al, 2021). The default settings of Seurat were used

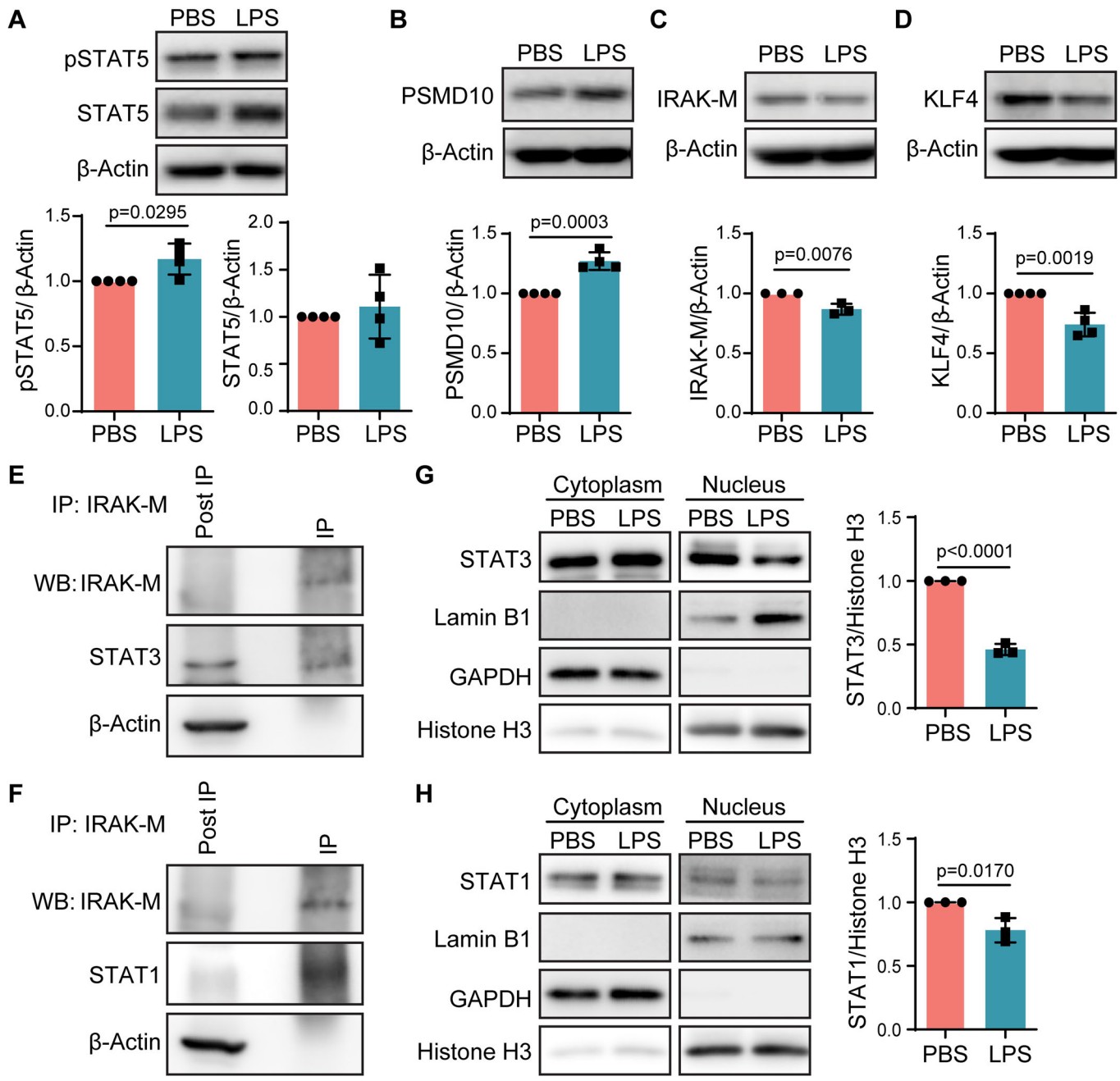

**Figure 7. Super-low dose LPS reprograms neutrophils through activating STAT5 and reducing IRAK-M.**

Purified neutrophils were treated with 100 pg/ml LPS or PBS as control overnight. Protein lysate were subjected to immunoblotting analysis. **(A)** Protein levels of pSTAT5 and STAT5, and relative expressions were normalized to β-actin. $n = 4$. **(B)** Protein levels and relative expressions of PSMD10. $n = 4$. **(C)** Protein levels and relative expressions of IRAK-M. $n = 3$. **(D)** Protein levels and relative expressions of KLF4. $n = 4$. **(E, F)** Co-immunoprecipitation analysis of IRAK-M with STAT3 and STAT1 from neutrophil lysates. **(G, H)** Protein levels of STAT3 and STAT1 in cytoplasm and nucleus. Quantification of nucleus STAT1/3 expressions normalized to histone H3. $n = 3$. Data information: Data **(A–D**, and **G, H)** are represented as means ± SD of independent biological replicates (*n*). Significance was calculated by unpaired two-tailed student *t* test. Source data are available online for this figure.

to perform quality control, normalization, and scaling of the data. Cells with more than 20% of reads from mitochondrial genes, and cells with more than 6500 or fewer than 200 unique genes were removed. Data were normalized and scaled before dimensionality reduction which were performed by principal component analysis (PCA), and UMAP were used for cell clustering. Marker genes that were differentially expressed in at least 10% of cells within a target cluster were obtained by using the non-parametric Wilcoxon rank sum test in R. Trajectory analysis was performed using the Monocle3 package in R as described (Lee et al, 2021).

## Histology and immunofluorescence analyses

Sections of colons were prepared as described before (Zhang et al, 2019). Paraffin-embedded sections were used for H&E staining (MilliporeSigma, catalog no. GHS132 and HT110132). The staining was viewed under Olympus stereomicroscope SZX7. For immunofluorescence staining, Paraffin-embedded sections were used for the detection of Ki67. Frozen sections were used for the detection of β-catenin, CD3, myeloperoxidase, and NK1.1. Briefly, after de-paraffinization (paraffin-embedded sections) or fixation with 4% paraformaldehyde for 15 min (frozen sections), sections were washed with PBS, then incubated with blocking buffer (10% goat serum/1%BSA/0.3%triton in PBS) for 1 h, followed by avidin/biotin blocking (Vectorlabs, catalog no. SP-2002). After sections were rinsed with PBS, primary antibodies were added, and incubated at 4 °C overnight. Then, biotinylated secondary antibodies were added, followed by fluorochrome-conjugated streptavidin. Between different antibodies, the sections were washed with PBS three times. The sections were mounted in Fluoromount-G® with DAPI (Southern Biotech, catalog no. 0100-20). Multiple viewing fields from each slide were captured under a Keyence fluorescence microscope. Images were quantified with BZ-X800 analyzer.

## Lamina propria cell isolation

Lamina propria cells were prepared as described previously (Zhang et al, 2019). Colons were flushed with ice-cold PBS, and cut into small pieces. A single-cell suspension was prepared using Lamina Propria Dissociation Kit (Miltenyi Biotec, catalog no. 130-097-410) and gentleMACS dissociator (Miltenyi Biotec). The cells were washed, passed through a 70-μm strainer, and resuspended in FACS buffer for further flow cytometry analyses.

## Flow cytometry

Leukocytes from bone marrow, spleen, and mesenteric lymph nodes were harvested as previously described (Zhang et al, 2019). For surface phenotype analyses, the single cells were stained with fluorescent-conjugated antibodies in the presence of Fc block (BD Biosciences, catalog no. 553142) in FACS buffer (1xHBSS supplemented with 2% FBS) for 20 min on ice. Propidium iodide (Thermofisher, catalog no. P3566) was also added before flow cytometry to determine the cell viability. Cytosol intracellular staining was performed using Cytofix/Cytoperm™ Plus fixation/permeabilization solution kit with BD GolgiStop™ (BD Biosciences, catalog no. 554715). Intracellular staining of Foxp3 and Ki67 was performed using Fixation/Permeabilization Solution Kit (ebioscience, catalog no. 00-5523). Antibodies are listed in Appendix Table S1. Stained cells were analyzed with a FACSCanto II (BD Biosciences). FACS plots shown were analyzed with FlowJo.

## T cell proliferation and activation assay

Primed neutrophils were co-cultured with splenic T cells as described before (Zhang et al, 2019). Briefly, splenic T cells were purified using EasySep™ Mouse T Cell Isolation Kit (Stem Cell, catalog no. 19851), according to the manufacturer's instruction. Purified T cells were labeled with 5,6-carboxyfluorescein diacetate succinimidyl (CFSE; Thermofisher, catalog no. C34554), according

to the manufacturer's instructions. CFSE-labeled T cells were mixed with primed neutrophils at a 1:1 ratio and co-cultured in anti-CD3 antibody (2 μg/ml; BioXcell, catalog no. BE0001-1) coated plates in the presence of anti-CD28 antibody (2 μg/ml; BioXcell, catalog no. BE0015-1). Conditioned medium was collected after 24-h coculture. CFSE signals and activation markers were analyzed by flow cytometry on gated CD4+ and CD8+ cells for 72-h coculture.

For human neutrophils, peripheral blood samples from healthy donors were purchased from Research Blood Components. Neutrophils were isolated using MojoSort™ whole blood human neutrophil isolation kit (BioLegend, catalog no. 480152). Human neutrophils were cultured in RPMI 1640 completed medium supplemented with human GM-CSF (1 ng/ml; Peprotech, catalog no. AF-300-03), and treated with LPS (100 pg/ml) or PBS as control for 24 h. For human T cells, peripheral blood was mixed with PBS (2%FBS) at 1:1 ratio and carefully loaded onto Ficoll® Paque Plus (MilliporeSigma, catalog no. GE17-1440-02), then centrifuged for 30 min at $800 \times g$. Peripheral blood mononuclear cells were collected and subjected to MojoSort™ human T cell isolation kit (BioLegend, catalog no. 480022). CFSE-labeled T cells were mixed with primed neutrophils at a 1:1 ratio and co-cultured in anti-CD3 antibody (1 μg/ml; BioLegend, catalog no. 317325) coated plates in the presence of anti-CD28 antibody (1 μg/ml; BioLegend catalog no. 302933) for 72 h. CFSE signals were analyzed by flow cytometry.

## NK cell proliferation assay

NK cells were purified from splenocytes using EasySep™ Mouse NK Cell Isolation Kit (Stem Cell, catalog no. 19855), according to the manufacturer's instruction. CFSE-labeled NK cells were mixed with primed neutrophils at 1:2 ratio and co-cultured in RPMI completed medium supplemented with IL-2 (25 ng/ml, Biolegend, catalog no. 714604) for 4 days. CFSE signals as well as NKG2A expression were analyzed by flow cytometry.

## NK cell killing assay

YAC-1 cells were purchased from ATCC (catalog no. TIB-160™). NK cells were purified from splenocytes and cultured in RPMI completed medium supplemented with IL-2 (25 ng/ml; Biolegend, catalog no. 714604) for 4 days. After wash, NK cells were co-cultured with neutrophils at 1:1 ratio. After 2-h priming in RPMI completed medium, the cells were transferred to an anti-NK1.1 (5 μg/ml; Biolegend, catalog no. 108702)-precoated plate for additional 4 h, then the supernatant was collected for ELISA analysis. Or after priming step, CFSE-labeled YAC-1 cells were introduced into the coculture system for 4 h. The quantification of live fluorescent-labeled YAC-1 cells was carried out by the addition of Bright Absolute Counting Beads (Life Technologies, catalog no. C36950) followed by flow cytometry analysis. In addition, lactate dehydrogenase (LDH) release in conditioned medium was measured using a colorimetric Lactate Dehydrogenase Assay kit (Abcam; catalog no. ab102526).

## Immunoblotting

Primed neutrophils were lysed in SDS lysis buffer (1%SDS, 10% glycerol, 0.05 M Tris-HCl pH6.8) with phosphatase and protease

inhibitor cocktails (MilliporeSigma, catalog no. P8340, P0044, P5726). Protein lysis was applied to SDS–PAGE and transferred to a polyvinylidene difluoride membrane. The membrane was blocked and probed with primary antibodies, followed by horseradish peroxidase-linked secondary antibodies and chemiluminescence ECL detection kit (Advansta, catalog no. K-12045-D50). Antibodies are listed in Appendix Table S1. Chemiluminescence signals were detected using the Fujifilm Intelligent Dark Box. Immunoblots were quantified using ImageJ.

Cell fractionation into cytoplasmic and nuclear components was achieved by using an NE-PER™ Nuclear and Cytoplasmic Extraction Kit (ThermoFisher, catalog no. 78833) according to the manufacturer's instructions. Subsequently, immunoblotting assay was conducted for these extracted nuclear and cytoplasmic fractions.

## Immunoprecipitation assay

WT bone marrow neutrophils were isolated and lysed in ice-cold Nonidet P40 lysis buffer supplemented with a complete protease inhibitor cocktail (MilliporeSigma, catalog no. P8340, P0044, P5726). The whole-cell lysates were utilized for immunoprecipitation with the anti-IRAK-M antibody following the manufacturer's instructions, supported by Pierce™ Classic Magnetic IP/Co-IP Kit (ThermoFisher, catalog no. 88804). Typically, 10 μg of IRAKM antibody was added to 500 μl of cell lysate, and the mixture was incubated at 4 °C overnight. Following this, Protein A/G-agarose beads were added, and the incubation was continued for an additional 1 h. Immunoprecipitates were thoroughly washed with lysis buffer and eluted with SDS loading buffer by boiling for 5 min. Both immunoprecipitated proteins and proteins in elution buffer were subjected to immunoblotting assay.

## ELISA

The levels of IFNγ (R&D System, catalog no. MIF00-1) and granzyme B (ThermoFisher, catalog no. BMS6029) in colon lysate and conditioned medium were measured using ELISA kits, according to the manufacturer's instructions. To prepare tissue lysate, colons were cut into small pieces and lysed in T-PER™ tissue protein extraction reagent (ThermoFisher, catalog no. 78510) with phosphatase and protease inhibitor cocktails by sonication. Cytokines in the colon were normalized by total protein weight.

## Neutrophil viability assay

Neutrophils were isolated from WT bone marrow cells by percoll gradient, as described before (Zhang et al, 2020). Neutrophils were cultured in RPMI completed Medium (RPMI 1640 medium supplemented with 10% fetal bovine serum, 2 mM L-glutamine, 10 mM HEPES, 1% penicillin/streptomycin) with or without GM-CSF (1 ng/mL), and with or without LPS (100 pg/mL) for 24 h. Then neutrophils were collected for cell counts, compared with the initial cell counts. Then, cultured cells subjected to apoptotic analysis by using an Annexin V Apoptosis Detection Kit APC (ThermoFisher, catalog no. C34554). Stained cells were analyzed with a FACSCanto II (BD Biosciences). FACS plots shown were analyzed with FlowJo.

### The paper explained

**Problem**

Neutrophils play key roles in modulating host immune environment, with both tumor-promoting and inhibiting effects. However, mechanisms for neutrophil polarization into either tumor-promoting or tumor-inhibiting states have not been well understood or examined.

**Results**

In this study we characterized the reprogramming of neutrophils by subclinical endotoxin into an immune-enhancing state with reduced pathogenic inflammation, characterized by CD177$^{lo}$CD11b$^{lo}$CD80$^{hi}$. Immune-enhancing neutrophils trained by low-dose LPS alleviate immune suppression in vitro and in vivo. When transfused into tumor-bearing recipient mice, trained neutrophils can reduce tissue inflammation and experimental colon tumor progression.

**Impact**

Our data provide novel insights into the polarization of neutrophils conducive for enhancing anti-tumor immunity and suggest that reprogramming immune-enhancing neutrophils may hold potential promise for novel anti-tumor treatment.

## In vivo neutrophil tracking

Bone marrow neutrophils from WT mice were purified using EasySep™ Mouse Neutrophil Enrichment Kit, according to the manufacturer's instruction. Purified neutrophils were labeled with CFSE at a concentration of 5 μm for 10 min at 37 °C. After WT mice were changed with DSS for 5 days, CFSE-labeled neutrophils were injected intravenously. One day after transfusion, mice were sacrificed and tissues were harvested for flow cytometry analysis.

## Immunohistochemistry staining

Frozen sections of colon tissues were used for the detection of CD3 and MPO. Sections (10 μm) were fixed in 4% neutral buffered formalin for 10 min, and stained with rabbit anti-MPO primary antibodies (Abcam, catalog no. ab9535) followed by a biotinylated anti-rabbit IgG antibody, ABC-HRP kit (Vectorlabs, catalog no. PK-6100), and DAB substrate kit (Vectorlabs, catalog no. SK-4100) according to the manufactory instructions. Then, the slides were further blocked and stained with rat anti-CD3 primary antibodies (Abcam, catalog no. 11089), followed by ImmPRESS®-AP goat anti-rat IgG (Mouse Adsorbed) polymer detection Kit (Vectorlabs, catalog no. MP-5444) and red substrate kit (Vectorlabs, catalog no. SK-5105). Slides were visualized under an Olympus microscope.

## Statistical analysis

All experiments were performed at least 3 times. Representative and reproducible results were shown. Mice were randomly assigned to each group of treatment. There were no inclusion/exclusion criteria. Statistical analysis was performed with Prism GraphPad Software 10.2.0. Values were expressed as means ± SD. The significance of the differences between two groups was assessed by two-tailed student $t$ test; comparison of multiple groups was

performed using ANOVA with multiple comparisons. $P < 0.05$ was considered statistically significant.

## For more information

Information about the research team and related works can be found at the authors' website https://li.biol.vt.edu.

## Data availability

The scRNAseq data sets are available in the following database: scRNAseq data: Gene Expression Omnibus GSE230237 (https://www.ncbi.nlm.nih.gov/geo/query/acc.cgi?acc=GSE230237).

The source data of this paper are collected in the following database record: biostudies:S-SCDT-10_1038-S44321-024-00100-7.

## Peer review information

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

## Acknowledgements

We appreciate the technical support from the Li Lab members such as Feng Xu during this study. This work was funded in part by National Institutes of Health grant R01 AI136386 and AI172133 to LL.

## Author contributions

**Yao Zhang**: Conceptualization; Data curation; Formal analysis; Validation; Investigation; Visualization; Methodology; Writing—original draft; Writing—review and editing. **Christina Lee**: Data curation; Formal analysis; Investigation; Methodology. **Shuo Geng**: Data curation; Investigation. **Jing Wang**: Data curation; Formal analysis; Investigation. **Udipta Bohara**: Software; Formal analysis. **Jacqueline Hou**: Data curation; Investigation; Writing—review and editing. **Ziyue Yi**: Software; Formal analysis. **Liwu Li**: Conceptualization; Supervision; Funding acquisition; Validation; Writing—original draft; Project administration; Writing—review and editing.

Source data underlying figure panels in this paper may have individual authorship assigned. Where available, figure panel/source data authorship is listed in the following database record: biostudies:S-SCDT-10_1038-S44321-024-00100-7.

## Disclosure and competing interests statement

The authors declare no competing interests.

