## [Peer Review File · EMBO Molecular Medicine]

Immune-enhancing neutrophils reprogrammed by subclinical low-dose endotoxin in cancer treatment

Yao Zhang, Christina Lee, Shuo Geng, Jing Wang, Udipta Bohara, Jacqueline Hou, Ziyue Yi, and Liwu Li

Corresponding author: Liwu Li (lwli@vt.edu)

Review Timeline:

Submission Date:	24th Oct 23
Editorial Decision:	21st Nov 23
Revision Received:	17th May 24
Editorial Decision:	14th Jun 24
Revision Received:	18th Jun 24
Accepted:	1st Jul 24

Editor: Poonam Bheda

Transaction Report:

21st Nov 2023

Dear Dr. Li,

Thank you for the submission of your manuscript to EMBO Molecular Medicine. We have now received feedback from the three reviewers who agreed to evaluate your manuscript. As you will see from the reports below, the referees acknowledge the interest of the study though they raise important concerns on your work and suggest that more experimental validation is needed to fully support the conclusions. In particular, they suggest:

- ensuring that the effects observed are due to LPS as GM-CSF also results in neutrophil activation
- to provide further evidence that this is specifically mediated by the neutrophils and not residual LPS
- analyzing whether there are any effects on neutrophil half-life with treatment of LPS or when transferred in vivo
- further validation of IRAK-M and KLF-4 roles
- addition of a control group for free growth of tumor for animal experiments

Addressing the reviewers' concerns in full will be necessary for further considering the manuscript in our journal, and acceptance of the manuscript will entail a second round of review. EMBO Molecular Medicine encourages a single round of revision only and therefore, acceptance or rejection of the manuscript will depend on the completeness of your responses included in the next, final version of the manuscript. For this reason, and to save you from any frustrations in the end, I would strongly advise against returning an incomplete revision.

We are expecting your revised manuscript within three months, if you anticipate any delay, please contact us.

We require:

4) A .docx formatted letter INCLUDING the reviewers' reports and your detailed point-by-point responses to their comments. As part of the EMBO Press transparent editorial process, the point-by-point response is part of the Review Process File (RPF), which will be published alongside your paper.

5) A complete author checklist, which you can download from our author guidelines (<https://www.embopress.org/page/journal/17574684/authorguide#submissionofrevisions>). Please insert information in the checklist that is also reflected in the manuscript. The completed author checklist will also be part of the RPF.

6) Please note that all corresponding authors are required to supply an ORCID ID for their name upon submission of a revised manuscript.

7) It is mandatory to include a 'Data Availability' section after the Materials and Methods. Before submitting your revision, primary datasets produced in this study need to be deposited in an appropriate public database, and the accession numbers and database listed under 'Data Availability'. Please remember to provide a reviewer password if the datasets are not yet public (see <https://www.embopress.org/page/journal/17574684/authorguide#dataavailability>).

In case you have no data that requires deposition in a public database, please state so in this section. Note that the Data Availability Section is restricted to new primary data that are part of this study. This study includes no data deposited in external repositories.

8) For data quantification: please specify the name of the statistical test used to generate error bars and P values, the number (n) of independent experiments (specify technical or biological replicates) underlying each data point and the test used to

calculate p-values in each figure legend. The figure legends should contain a basic description of n, P and the test applied. Graphs must include a description of the bars and the error bars (s.d., s.e.m.). Please provide exact p values.

13) Author contributions: CRediT has replaced the traditional author contributions section because it offers a systematic machine readable author contributions format that allows for more effective research assessment. Please remove the Authors Contributions from the manuscript and use the free text boxes beneath each contributing author's name in our system to add specific details on the author's contribution. More information is available in our guide to authors.

Please also suggest a striking image or visual abstract to illustrate your article as a PNG file 550 px wide x 300-600 px high. Share synopsis text and image, as well as eTOC:

Please note that these would be the final versions and changes during proofing are usually not allowed

16) As part of the EMBO Publications transparent editorial process initiative (see our Editorial at <http://embomolmed.embopress.org/content/2/9/329>), EMBO Molecular Medicine will publish online a Review Process File (RPF) to accompany accepted manuscripts.

In the event of acceptance, this file will be published in conjunction with your paper and will include the anonymous referee reports, your point-by-point response and all pertinent correspondence relating to the manuscript. Let us know whether you agree with the publication of the RPF and as here, if you want to remove or not any figures from it prior to publication. Please note that the Authors checklist will be published at the end of the RPF.

I look forward to receiving your revised manuscript.

Yours sincerely,

Poonam Bheda

Poonam Bheda, PhD
Scientific Editor
EMBO Molecular Medicine

***** Reviewer's comments *****

Referee #1 (Comments on Novelty/Model System for Author):

In this study, the authors propose that neutrophils, when trained with super-low doses of endotoxin, acquire a robust immune-enhancing phenotype. This phenotype is characterized by a diminished suppression of adaptive T cells, which is facilitated through the activation of STAT5. The authors use single-cell RNA sequencing data to demonstrate that neutrophils stimulated by a super-low dose of endotoxin exhibit a CD177^{lo}CD11b^{lo}CD80^{hi}CD40^{hi}Dectin2^{hi} phenotype. The capacity of these activated neutrophils to inhibit tumor growth adds a significant understanding to the "Coley's toxin" concept, which is particularly engaging given the high plasticity of tumor-associated neutrophils. Recent publications have highlighted the anti-tumor functions following neutrophil reprogramming, placing this manuscript as a valuable contribution to the field. Nonetheless, there are several areas of concern that need to be addressed:

In Figure 1B, the neutrophils treated overnight with super-low dose LPS plus GM-CSF form a distinct CD177^{lo} cluster with altered expression of genes associated with both immunosuppression and immune enhancement. The treated population deviates from the original CD177^{lo} cluster. The authors need to clarify these findings more explicitly.

Fig 1. Need to include FACS plots.

The manuscript can be strengthened by additional analyses, such as trajectory studies, to determine the precursor population of the post-treatment neutrophils.

Given that both LPS and GM-CSF can influence neutrophil function, the authors should provide more evidence to support their claim that the anti-tumor function is primarily attributed to LPS, as suggested in the manuscript.

Appendix Figure S1 indicates an upregulation of PDL1 (encoded by CD274) following low-dose LPS treatment. The authors should explain this observation and its implications.

In Figure 2c, there seems to be a convergence in weight change and stool clinical score between the two groups. Does this imply that the difference in tumor burden is minimal at the early time points? If so, when does the disparity become apparent?

Figure 3 aims to show that the transfusion of immune-enhancing neutrophils increases T cell activity. To convincingly demonstrate this, the authors should analyze not only the quantity but also the functional state of CD4⁺ and CD8⁺ T cells at the tumor site in the colon, in addition to the spleen and lymph nodes.

The manuscript posits that STAT5 and IRAK-M play roles in reducing the immune suppressive abilities of neutrophils. However, the presented results appear to show minimal differences. This aspect needs strengthening to substantiate the claims.

Figure 5. Reprogrammed neutrophils by super low-dose LPS increase T cell proliferation and activation in vitro. More accurately, this treatment abolished the neutrophil immunosuppression activity. It did not increase T cell proliferation and activation, compared to "no neutrophils".

Figure 6. Similarly, LPS-treated neutrophils still decreased, instead of increased, NK cell proliferation and functional ability, compared to "no neutrophils".

Referee #1 (Remarks for Author):

In this study, the authors propose that neutrophils, when trained with super-low doses of endotoxin, acquire a robust immune-enhancing phenotype. This phenotype is characterized by a diminished suppression of adaptive T cells, which is facilitated through the activation of STAT5. The authors use single-cell RNA sequencing data to demonstrate that neutrophils stimulated by a super-low dose of endotoxin exhibit a CD177^{lo}CD11b^{lo}CD80^{hi}CD40^{hi}Dectin2^{hi} phenotype. The capacity of these activated neutrophils to inhibit tumor growth adds a significant understanding to the "Coley's toxin" concept, which is particularly engaging given the high plasticity of tumor-associated neutrophils. Recent publications have highlighted the anti-tumor functions following neutrophil reprogramming, placing this manuscript as a valuable contribution to the field. Nonetheless, there are several areas of concern that need to be addressed:

In Figure 1B, the neutrophils treated overnight with super-low dose LPS plus GM-CSF form a distinct CD177^{lo} cluster with altered expression of genes associated with both immunosuppression and immune enhancement. The treated population deviates from the original CD177^{lo} cluster. The authors need to clarify these findings more explicitly.

Fig 1. Need to include FACS plots.

The manuscript can be strengthened by additional analyses, such as trajectory studies, to determine the precursor population of the post-treatment neutrophils.

Given that both LPS and GM-CSF can influence neutrophil function, the authors should provide more evidence to support their claim that the anti-tumor function is primarily attributed to LPS, as suggested in the manuscript.

Appendix Figure S1 indicates an upregulation of PDL1 (encoded by CD274) following low-dose LPS treatment. The authors should explain this observation and its implications.

In Figure 2c, there seems to be a convergence in weight change and stool clinical score between the two groups. Does this imply that the difference in tumor burden is minimal at the early time points? If so, when does the disparity become apparent?

Figure 3 aims to show that the transfusion of immune-enhancing neutrophils increases T cell activity. To convincingly demonstrate this, the authors should analyze not only the quantity but also the functional state of CD4⁺ and CD8⁺ T cells at the tumor site in the colon, in addition to the spleen and lymph nodes.

The manuscript posits that STAT5 and IRAK-M play roles in reducing the immune suppressive abilities of neutrophils. However, the presented results appear to show minimal differences. This aspect needs strengthening to substantiate the claims.

Figure 5. Reprogrammed neutrophils by super low-dose LPS increase T cell proliferation and activation in vitro. More accurately, this treatment abolished the neutrophil immunosuppression activity. It did not increase T cell proliferation and activation, compared to "no neutrophils".

Figure 6. Similarly, LPS-treated neutrophils still decreased, instead of increased, NK cell proliferation and functional ability, compared to "no neutrophils".

Referee #2 (Comments on Novelty/Model System for Author):

Please refer to the detail comments in the remarks below.

Referee #2 (Remarks for Author):

I enjoyed reading the manuscript by Zhang et al. exploring the immune-enhancing effects induced by low-dose LPS on neutrophils. The authors conducted a thorough evaluation of low-dose LPS-primed neutrophils in a DSS-AOM colon cancer model, and further elucidated the mechanisms underlying tumor suppression through various in vitro assays. Their findings shed light on the potential workings of Coley's toxin, particularly in terms of modulating neutrophils, perhaps by downregulating IRAK-M and suppressing T cell and NK cell activities. However, I have a few suggestions for the authors to consider in order to enhance the clarity and depth of the manuscript:

1. The authors showed the upregulation of immune-enhancing genes such as CD44, CD80, CD40, EHD1, and Dectin2 through low-dose LPS priming (Fig1). However, the UMAP analysis in Fig1A clearly indicates that LPS-primed neutrophils differ from those treated with PBS, at least at the transcriptional level. Since neutrophils have a short lifespan, it would be valuable for the authors to investigate whether LPS priming alters the half-life of neutrophils or influences the expression of genes related to their survival and proliferation.

2. Building upon the previous query, the authors might consider examining whether transferred neutrophils exhibit varying half-lives in vivo through the use of CD45.1 or CFSE labeling of AT neutrophils. This investigation could provide insights into the

exact location where AT neutrophils carry out their functions in T cell and NK cell activation. Moreover, it would be pertinent to determine if AT neutrophils can be found within the polyps or elsewhere.

3. Given LPS is notorious to be removed from the culture, it is crucial to rule out the possibility that the observed in vivo antitumor effects are mediated by neutrophils and not by trace amounts of transferred endotoxin. The authors should include strategies to account for and mitigate this potential confounding factor.

undefined. The downregulation of IRAK-M and KLF-4 is an intriguing finding. However, the manuscript could benefit from a clearer establishment of the causal relationship. Implementing inhibitor or knockout models could effectively elucidate the specific roles of these factors in the observed phenomenon.

Referee #3 (Comments on Novelty/Model System for Author):

The grouping of animal experiments is quite flawed, there is a lack of control group for free growth of tumor.

Referee #3 (Remarks for Author):

The article is meant to be novel--training neutrophils by super-low dose LPS to enhance immunity for anti-tumor treatment. But the experimental design is rough and vulnerable:

1. The neutrophil subsets according to CD177 expression in Figure 1 are independent of full text content.
2. The grouping of animal experiments is quite flawed, there is a lack of control group for free growth of tumor. What is the significance of two treatment groups for comparison?
3. How was the super low dose LPS concentration determined? Are there any references or pre-experiments? Why should GM-CSF be added to LPS primed neutrophils? How is it determined whether super low dose LPS or GM-CSF plays a role? Where were the GM-CSF single positive group control data?
4. Figure 3 and figure 4 lack data of T cells and NK cells in tumors.
5. How does the dosage and frequency of neutrophil infusion determine? Are there any references or pre-experiments?
6. What is the ratio of in vitro neutrophil coculture with T cells and NK cells determined?
7. The data and conclusions of T cell proliferation in vitro co-culture assays are debatable.
8. The full texts were histogram data, without any corroborating data such as morphology and gross view of tissues.
9. Writing error: please unify the statistical expression " * " and "**".

May 5th 2024

Dear Editor,

We are pleased to hear many positive comments and appreciation for our manuscript, and truly grateful for the very thoughtful suggestions to further improve the clarity of our novel study. We addressed each of these comments carefully either through additional experiments or through careful reasoning of the experimental systems based on independent studies. Manuscripts have been revised accordingly, with revision responses detailed below:

Referee #1:

In this study, the authors propose that neutrophils, when trained with super-low doses of endotoxin, acquire a robust immune-enhancing phenotype. This phenotype is characterized by a diminished suppression of adaptive T cells, which is facilitated through the activation of STAT5. The authors use single-cell RNA sequencing data to demonstrate that neutrophils stimulated by a super-low dose of endotoxin exhibit a CD177^{lo}CD11b^{lo}CD80^{hi}CD40^{hi}Dectin2^{hi} phenotype. The capacity of these activated neutrophils to inhibit tumor growth adds a significant understanding to the "Coley's toxin" concept, which is particularly engaging given the high plasticity of tumor-associated neutrophils. Recent publications have highlighted the anti-tumor functions following neutrophil reprogramming, placing this manuscript as a valuable contribution to the field. Nonetheless, there are several areas of concern that need to be addressed:

We appreciate the overall summary and positive comments for the reviewer.

In Figure 1B, the neutrophils treated overnight with super-low dose LPS plus GM-CSF form a distinct CD177^{lo} cluster with altered expression of genes associated with both immunosuppression and immune enhancement. The treated population deviates from the original CD177^{lo} cluster. The authors need to clarify these findings more explicitly.

This is a very insightful comment and we are also keenly aware of this. Indeed, upon low-dose LPS training, a distinct CD177^{lo} population emerges. As shown in the representative gene expression profiles, the overall CD177^{lo} populations exhibit reduced pathogenic inflammatory features. Upon LPS training, the CD177^{lo} population gained additional immune-enhancing characteristics such as elevated expression of CD80, CD86, Dectin1, and EHD1. Therefore, low-dose LPS training will preferentially give rise to immune-enhancing neutrophils without excessive pathogenic inflammation. We have discussed these unique aspects in the revised manuscript.

Fig 1. Need to include FACS plots. The manuscript can be strengthened by additional analyses, such as trajectory studies, to determine the precursor population of the post-treatment neutrophils.

As suggested, we have added the FACS plots in the revised supplemental figure. We further performed trajectory analyses which provided an intriguing clue that there were two separate origins of neutrophils, with the CD177^{hi} population giving rise to the pathogenic inflammatory CD177^{intermediate} population, and the CD177^{lo} population giving rise to the less-pathogenic; yet immune-enhancing population with elevated CD80 expression following low-dose LPS challenge. These analyses have been included in the revised manuscript. We also noticed that the

Invent the Future

immune-enhancing CD177^{lo} population has highly elevated expression of Ly6a/e. While this manuscript was under revision, another independent study just published in Cancer Cell reviewed that both human and murine Ly6a/e^{hi} neutrophil subsets are the most potent subsets capable of enhancing anti-tumor immune defense, highly consistent with our current finding. We discussed this corroborative independent study in the revised manuscript.

Given that both LPS and GM-CSF can influence neutrophil function, the authors should provide more evidence to support their claim that the anti-tumor function is primarily attributed to LPS, as suggested in the manuscript.

As well-tested by independent teams, low-level of GM-CSF (in this case, 1ng/ml) is necessary for maintaining neutrophil survival during in vitro culture and can lead to robust maintenance of both human and murine neutrophils for the long-term in vitro culture purpose. Although higher levels of GM-CSF (>100 ng/ml) can lead to neutrophil activation, very low level of GM-CSF (<10 ng/ml) can selectively assist to maintain the survival of neutrophils. We provided additional validation that neutrophils maintained with low-level GM-CSF indeed is needed to sustain robust survival following overnight culture, and that LPS addition will not alter neutrophil survival.

All experiments were conducted comparing GM-CSF maintained neutrophils with or without LPS challenge. Therefore, effects we observed were due to LPS stimulation as compared to control neutrophils only maintained by low level GM-CSF.

We included these in the supplemental figures and the revised discussion.

Appendix Figure S1 indicates an upregulation of PDL1 (encoded by CD274) following low-dose LPS treatment. The authors should explain this observation and its implications.

Indeed, low-dose LPS can initiate the mild induction of PD-L1. The key point is that low-dose LPS retains the capability of potentially inducing the expression of immune-enhancing mediators. In contrast, higher dose LPS loses the capability of inducing immune-inducing mediators, while can only robustly induce PD-L1. The dichotomy of innate training by low-dose LPS is the novelty of this manuscript, and we discussed in more detail in the revised manuscript.

In Figure 2c, there seems to be a convergence in weight change and stool clinical score between the two groups. Does this imply that the difference in tumor burden is minimal at the early time points? If so, when does the disparity become apparent?

In this model of tumor induction, tumor burdens were minimal in the first 4-6 weeks. Therefore, similar studies with this model all tend to analyze the late stage of tumor burdens for a robust readout.

Figure 3 aims to show that the transfusion of immune-enhancing neutrophils increases T cell activity. To convincingly demonstrate this, the authors should analyze not only the quantity but also the functional state of CD4+ and CD8+ T cells at the tumor site in the colon, in addition to the spleen and lymph nodes.

This is a very good suggestion. We performed selected measurement of some key activation markers, and included in the revised manuscript. However, due to the scope of the work that focuses on defining innate neutrophil training by very low dose LPS, the functional consequences with the subsequent regulations of diverse adaptive immune cells such as T cells, and B cells cannot be fully defined in this current work. We stated our limitations of the current work in the revised discussion.

The manuscript posits that STAT5 and IRAK-M play roles in reducing the immune suppressive abilities of neutrophils. However, the presented results appear to show minimal differences. This aspect needs strengthening to substantiate the claims.

We performed additional studies to validate the underlying mechanisms. IRAK-M preferentially contributes to the induction of immune suppressive genes (CD11b/PD-L1) through activating STAT1/3 and its downstream factor KLF4, while suppressing immune-enhancing genes through suppressing STAT5. We previously reported the suppression of STAT5 by IRAK-M.

In this revision study, we focused on solidifying the mechanism for IRAK-M mediated activation of STAT3/1. Based on an independent study that IRAK-M can closely interact with STAT3 and potentially stabilize STAT3, we performed additional experiment testing the interaction of IRAK-M with both STAT1 and STAT3 in the setting of neutrophils. Indeed, through co-immunoprecipitation assays, we detected an interaction of IRAK-M with both STAT1 and STAT3 (revised Fig 7E and F). Further, we performed nuclear isolation and tested the nuclear levels of STAT1/3. As shown in revised Fig 7G and H, LPS priming significantly reduced STAT3 and STAT1 in the nuclear fraction, validating the reduction of nuclear levels of STAT3 and STAT1 by LPS. In addition to analyzing total cellular protein lysates, we harvested nuclear fractions and showed that super-low dose LPS selectively reduced nuclear levels of STAT3/1 responsible for the expression of KLF4 as well as downstream pathogenic inflammatory/immune suppressive mediators such as CD11B.

These data were added to the revised manuscript, together with proper discussion of limitations and future directions.

Figure 5. Reprogrammed neutrophils by super low-dose LPS increase T cell proliferation and activation in vitro. More accurately, this treatment abolished the neutrophil immunosuppression activity. It did not increase T cell proliferation and activation, compared to "no neutrophils".

Figure 6. Similarly, LPS-treated neutrophils still decreased, instead of increased, NK cell proliferation and functional ability, compared to "no neutrophils".

We re-worded our description as the reviewers thoughtfully pointed out, in that super-low dose LPS blocked the neutrophil immunosuppression activity.

Referee #2 (Remarks for Author):

I enjoyed reading the manuscript by Zhang et al. exploring the immune-enhancing effects induced by low-dose LPS on neutrophils. The authors conducted a thorough evaluation of low-dose LPS-primed neutrophils in a DSS-AOM colon cancer model, and further elucidated the mechanisms underlying tumor suppression through various in vitro assays. Their findings shed light on the potential workings of Coley's toxin, particularly in terms of modulating neutrophils, perhaps by downregulating IRAK-M and suppressing T cell and NK cell activities. However, I have a few suggestions for the authors to consider in order to enhance the clarity and depth of the manuscript:

We appreciate the overall summary and positive comments for the reviewer.

1. The authors showed the upregulation of immune-enhancing genes such as CD44, CD80, CD40, EHD1, and Dectin2 through low-dose LPS priming (Fig1). However, the UMAP analysis in Fig1A clearly indicates that LPS-primed neutrophils differ from those treated with PBS, at least at the transcriptional level. Since neutrophils have a short lifespan, it would be valuable for the authors to investigate whether LPS priming alters the half-life of neutrophils or influences the expression of genes related to their survival and proliferation.

2. Building upon the previous query, the authors might consider examining whether transferred neutrophils exhibit varying half-lives in vivo through the use of CD45.1 or CFSE labeling of AT neutrophils. This investigation could provide insights into the exact location where AT neutrophils carry out their functions in T cell and NK cell activation. Moreover, it would be pertinent to determine if AT neutrophils can be found within the polyps or elsewhere.

We appreciate these very thoughtful comments. We used the in vitro culture system supplemented with low-level GM-CSF, which was known to sustain the survival of neutrophils in vitro. We performed further validation and demonstrated that low-dose LPS does not impact neutrophil survival. We incorporated these into the supplemental data of the manuscript.

We have also tracked labeled neutrophils when injected in vivo, and demonstrated that indeed they can be naturally homed into the gut mucosal tissues.

3. Given LPS is notorious to be removed from the culture, it is crucial to rule out the possibility that the observed in vivo antitumor effects are mediated by neutrophils and not by trace amounts of transferred endotoxin. The authors should include strategies to account for and mitigate this potential confounding factor. *undefined*. The downregulation of IRAK-M and KLF-4 is an intriguing finding. However, the manuscript could benefit from a clearer establishment of the causal relationship. Implementing inhibitor or knockout models could effectively elucidate the specific roles of these factors in the observed phenomenon.

We acknowledge that the main focus of this work is to define the unique polarization of neutrophils by very-low dose LPS. The unique phenotype of neutrophils trained by low-dose LPS is evident through RNAseq analyses. The next phase of functional implication for in vivo treatment of tumor can also be seen with the re-programmed neutrophils. Once the neutrophils are reprogrammed in vitro, they were thoroughly washed with LPS free PBS before being introduced into the mice. The dosage we used for the in vitro challenge is in the pg/ml range. Such very low-dosage of LPS would be readily neutralized by plasma serum when administered in vivo, and would be below the range of current detection method. Nevertheless, we acknowledge such limitation in the discussion, in that our work provides an initial concept of neutrophil reprogramming, and future defined studies are needed to translate into therapeutic strategies.

Referee #3 (Remarks for Author):

The article is meant to be novel--training neutrophils by super-low dose LPS to enhance immunity for anti-tumor treatment. But the experimental design is rough and vulnerable:

1. *The neutrophil subsets according to CD177 expression in Figure 1 are independent of full text content.*

We appreciate the thoughtful comment. Indeed, as we clarified with the other reviewer, upon low-dose LPS training, a distinct CD177^{lo} population emerges. As shown in the representative gene expression profiles, the overall CD177^{lo} populations exhibit reduced pathogenic inflammatory features. Upon LPS training, the CD177^{lo} population gained additional immune-enhancing characteristics such as elevated expression of CD80, CD86, Dectin1, and CHD1. Therefore, low-dose LPS training will preferentially give rise to immune-enhancing neutrophils without excessive pathogenic inflammation. We have discussed these unique aspects in the revised manuscript.

2. *The grouping of animal experiments is quite flawed, there is a lack of control group for free growth of tumor. What is the significance of two treatment groups for comparison?*

We have employed this in vivo animal system widely and robustly in our previous studies, which serve as baseline controls for free growth of tumors. The tumor burdens of mice transfused with control neutrophils are well-within the range of tumor burdens of mice we previously reported for free tumor growth without any neutrophil transfer. We did such comparison and included in the revised manuscript. In light of the 3 R principle of conducting animal research (*Replacement, Reduction and Refinement*) mandated by our institution following the NIH guideline, we restrain our current study to focus on varying only one variable among the treatment groups (tumor-mice receiving neutrophils programmed without or with LPS).

3. How was the super low dose LPS concentration determined? Are there any references or pre-experiments? Why should GM-CSF be added to LPS primed neutrophils? How is it determined whether super low dose LPS or GM-CSF plays a role? Where were the GM-CSF single positive group control data?

Likewise, we universally applied very-low dose GM-CSF for the well-known purpose of sustaining the robust survival of cultured neutrophils. As independently reported by others, although higher levels of GM-CSF (>100 ng/ml) can lead to neutrophil activation, very low level of GM-CSF (<10 ng/ml) can selectively assist to maintain the survival of neutrophils. As well-tested by independent teams, low-level of GM-CSF (in this case, 1ng/ml) is necessary for maintaining neutrophil survival during in vitro culture and can lead to robust maintenance of both human and murine neutrophils for the long-term in vitro culture purpose. We provided additional validation that neutrophils maintained with low-level GM-CSF indeed is needed to sustain robust survival following overnight culture, and that LPS addition will not alter neutrophil survival.

The super-low dose LPS concentration used for the in vitro programming was based on our previously published studies, showing that 5-100 pg/ml LPS was within the threshold of inducing immune enhancing effects without triggering exhaustion or tolerance.

4. Figure 3 and figure 4 lack data of T cells and NK cells in tumors.

Indeed, we performed some analyses of T cells and NK cells within tumor environment, and provided these data in the revised supplemental figure S7. In the meantime, we acknowledge the limited scope of our study, and discussed future need for comprehensive studies.

5. How does the dosage and frequency of neutrophil infusion determine? Are there any references or pre-experiments?

For the in vivo assays, we adopted our optimization based on our previous related studies. We cited these previous studies in the revised manuscript.

6. What is the ratio of in vitro neutrophil coculture with T cells and NK cells determined?

Likewise, we performed extensive in vitro pilot studies in the past, and based on our optimization from previous published studies by others as well as our related studies. We cited these previous studies in the revised manuscript.

7. The data and conclusions of T cell proliferation in vitro co-culture assays are debatable.

We acknowledge the limitation, and carefully tuned down our conclusion in the revised manuscript. We would like to emphasize that this study is a test of principle for neutrophil reprogramming dynamics. Future systems in vivo studies must be performed, based on this conceptual principle, to further optimize potential translational in vivo studies.

8. *The full texts were histogram data, without any corroborating data such as morphology and gross view of tissues.*

We appreciate the suggestion, and performed additional immune-histological works, and provided relevant tissue images to further complement our existing data. These additional data were incorporated into the revised manuscript.

9. *Writing error: please unify the statistical expression " * " and "**".*

We have revised accordingly as suggested.

Together, we greatly appreciate the very thoughtful comments from all reviewers, which helped us to improve the readability and clarification of this work.

Sincerely Yours,

Liwu Li, Ph.D., FAHA, FAAAS
Endowed Professor of Immunology and Inflammation Biology
Virginia Tech
Immediate Past-President, Inflammation Research Association

14th Jun 2024

Dear Dr. Li,

Thank you for the submission of your revised manuscript to EMBO Molecular Medicine. Your manuscript has now been re-reviewed by two of the original reviewers, both of whom are supportive of your manuscript. While unfortunately Reviewer 3 was not available to re-review the manuscript, in a cross-commenting session with Reviewers 1 and 2, one of the reviewers also provided their impression that Reviewer 3 previously had valid concerns, some of which were not fully addressed (e.g. the experiments related to T cell proliferation), but that overall the manuscript was still suitable for publication in EMBO Molecular Medicine. Therefore, based on their advice, I am pleased to inform you that we will be able to accept your manuscript pending the following final amendments:

1) Please format the Data availability section describing how the data, code etc. have been made available according to the following example:

"The datasets and computer code produced in this study are available in the following databases:

- Chip-Seq data: Gene Expression Omnibus GSE46748 (<https://www.ncbi.nlm.nih.gov/geo/query/acc.cgi?acc=GSE46748>)
- [data type]: [full name of the resource] [accession number/identifier] ([doi or URL or identifiers.org/DATABASE:ACCESSION])"

2) Please rename "Competing Interests" to "Disclosure and competing interests statement". We updated our journal's competing interests policy in January 2022 and request authors to consider both actual and perceived competing interests. Please review the policy <https://www.embopress.org/competing-interests> and update your competing interests if necessary.

3) The "Consent for publication" statement is not necessary and should be removed from the main manuscript.

4) Author contributions: Please remove it from the manuscript and specify author contributions in our submission system.

CRedit has replaced the traditional author contributions section because it offers a systematic machine-readable author contributions format that allows for more effective research assessment. You are encouraged to use the free text boxes beneath each contributing author's name to add specific details on the author's contribution. More information is available in our guide to authors:

<https://www.embopress.org/page/journal/17574684/authorguide#authorshipguidelines>

5) In the Materials and Methods, please take care of the following:

- The use of human samples requires information on the authority granting ethics approval (e.g. IRB) and informed consent. If the need for approval is waived, please cite the reason (e.g. non-human subject research because the samples used were de-identified/coded with no identifying information) and legislation in the relevant methods section.

- Animals: Please move the ethics statement and the approval committee for research on animals into the section where animal experiments are described in the Materials and Methods. Please also ensure that sex of the mice is also reported.

- Cell lines: Please include all information requested in the author checklist for cell lines used in the manuscript (accession number in repository or supplier name, catalog number, clone number, and/or RRID). Please also be sure to include a sentence in the Materials and Methods as to whether or not the cell lines were recently authenticated and tested for mycoplasma contamination.

6) Please place individual sections of the manuscript in the following order: Title page - Abstract & Keywords - Introduction - Results - Discussion - Materials & Methods - Data Availability - Acknowledgements - Disclosure and Competing Interests Statement - The Paper Explained - For More Information - References - Figure Legends - Expanded View Figure Legends.

7) For the figures and figure legends, please take care of the following:

- Please note that the legends for figures 4d-g is not provided in the sequential manner (legend for figures 4e, g is provided before legend of figures 4d, f, respectively). This needs to be rectified.

- Please note that the exact p values are not provided in the legends of figures 1c; 2c-d, f-g; 7g.

8) Appendix file: In the Appendix file, please remove the Supplementary Methods and add these to the main manuscript. In addition, the title on page 12 for "Supplementary Table 1: needs correcting to "Appendix Table S1". Finally, please remove any highlighting from the text, as this file is uploaded as a PDF as is.

9) Funding: Please note that funding information should be given in the "Acknowledgements" section (not in its own separate section).

10) Synopsis:

- Synopsis image: The synopsis image should have the dimensions of 550 pixels wide x (250-400) pixels high. When we resize the image to 550 pixels wide, the image is not sufficiently high.

11) Source Data: Typically we suggest that FACS data are provided as '.fcs' or other standard FACS file formats. We would suggest that you include the files as Source Data for the relevant figures.

12) The Paper Explained: Please add "The Paper Explained" to the main manuscript text (not uploaded as a separate document).

13) For more information: This space should be used to list relevant web links for further consultation by our readers. Could you identify some relevant ones and provide such information as well? Some examples are patient associations, relevant databases, OMIM/proteins/genes links, author's websites, etc...

- 14) As part of the EMBO Publications transparent editorial process initiative (see our policy here: https://www.embopress.org/transparent-process#Review_Process), EMBO Molecular Medicine will publish online a Peer Review File (PRF) to accompany accepted manuscripts. This file will be published in conjunction with your paper and will include the anonymous referee reports, your point-by-point response and all pertinent correspondence relating to the manuscript. Let us know whether you agree with the publication of the PRF and as here, if you want to remove or not any figures from it prior to publication. Please note that the Authors checklist will be published at the end of the PRF.
- 15) Please provide a point-by-point letter including my above requests and your detailed responses (as Word file).

I look forward to reading a new revised version of your manuscript as soon as possible.

Yours sincerely,

Poonam Bheda

Poonam Bheda, PhD
Scientific Editor
EMBO Molecular Medicine

***** Reviewer's comments *****

Referee #1 (Remarks for Author):

The revised manuscript has been significantly improved by the additional data. The authors have made a forthright effort to address the criticisms raised in the previous review. I am happy to recommend this for EMBO Molecular Medicine without further revision. This study reveals a novel mechanism and provides an attractive clinical approach to reprogramming innate neutrophils into an immune-enhancing state conducive for the treatment of cancer.

Referee #2 (Comments on Novelty/Model System for Author):

This work demonstrates a promising approach to using reprogrammed neutrophils with anti-tumor translational potential. Given the short lifespan of neutrophils, this immune cell-mediated therapy, which does not require genetic modification, is considered safer than current genetically modified methods such as CAR-based approaches.

Referee #2 (Remarks for Author):

The authors have fully addressed my concerns.

The authors addressed the minor editorial issues.

1st Jul 2024

Dear Dr. Li,

We are pleased to inform you that your manuscript is accepted for publication and is now being sent to our publisher to be included in the next available issue of EMBO Molecular Medicine.

Yours sincerely,

Poonam Bheda, PhD
Scientific Editor
EMBO Molecular Medicine
